# Graph-Skeleton: Less than 2% Nodes are Sufficient to Represent Billion-Scale Graph

## ABSTRACT

Due to the ubiquity of graph data on the web, web graph mining has become a hot research spot. Nonetheless, the prevalence of large-scale web graphs in real applications poses significant challenges to storage, computational capacity and graph model design. Despite numerous studies to enhance the scalability of graph models, a noticeable gap remains between academic research and practical web graph mining applications. One major cause is that in most industrial scenarios, only a small part of nodes in a web graph are actually required to be analyzed, where we term these nodes as *target nodes*, while others as *background nodes*. In this paper, we argue that properly fetching and condensing the background nodes from massive web graph data might be a more economical shortcut to tackle the obstacles fundamentally. To this end, we make the first attempt to study the problem of massive background nodes compression for target nodes classification. Through extensive experiments, we reveal two critical roles played by the background nodes in target node classification: enhancing *structural connectivity* between target nodes, and *feature correlation* with target nodes. Following this, we propose a novel *Graph-Skeleton* model, which properly fetches the background nodes, and further condenses the semantic and topological information of background nodes within similar target-background local structures. Extensive experiments on various web graph datasets demonstrate the effectiveness and efficiency of the proposed method. In particular, for MAG240M dataset with 0.24 billion nodes, our generated skeleton graph achieves highly comparable performance while only containing 1.8% nodes of the original graph.

## CCS CONCEPTS

• **Information systems** → **Data mining**; **Data compression**.

## KEYWORDS

Graph Mining, Data Compression, Large-Scale Web Graph, Graph Neural Networks

ACM Reference Format:
Anonymous Author(s). 2024. Graph-Skeleton: Less than 2% Nodes are Sufficient to Represent Billion-Scale Graph. In *Proceedings of the ACM Web Conference 2024 (WWW '24), MAY 13 - 17, Singapore*. ACM, New York, NY, USA, 19 pages. https://doi.org/XXXXXXX.XXXXXXX

## 1 INTRODUCTION

The ubiquity of graph data, especially in the form of web graphs, has made web graph mining a hot research topic. These web graphs are crucial for various applications, including web search [6, 15], social network analysis [42, 43], recommendation systems [31, 34], and more. However, it remains a big challenge to deploy the graph models on large-scale web graphs. In practice, web graph data can be extremely large [45]. Take Facebook for instance, there are over 2.93 billion monthly active users [9]. Despite remarkable progress such as node sampling [4, 16, 45], model simplification [35, 38] to enhance the scalability of graph models on large-scale web graph mining tasks, it remains a large gap between academic research and practical web applications. One major reason is that in most industrial scenarios, not all nodes in a web graph are actually required to be analyzed [21]. We take DGraph, a real-world financial & social network dataset (users as nodes, social relationships between users as edges) [21], as an example to further illustrate this. Given a fraudster identification task among loan users, only the users with loan records needs to be classified, while the other users without loan behavior do not. Under this circumstance, we term the loan users as *target nodes*, while the others as *background nodes*. This target & background property is prevalent in web graph mining scenarios. Moreover, the number of background nodes is typically much larger than that of target nodes. For instance, to predict papers' subject areas in MAG240M [19], only 1.4 million Arxiv papers are concerned with classification among 240 million nodes.

Intuitively, it may not be an economical solution to deploy complex graphical models on massive web data just for a small number of node classification. It can pose significant challenges in terms of time and memory costs, as well as model design. Alternatively, a proper method to fetch and condense the useful background nodes from massive web data might be a shortcut to fundamentally tackle the above obstacles. Nevertheless, how to fetch and condense the informative background nodes remains an open question at present. Background nodes can play diverse roles in target nodes classification, yet there is little understanding of how the background nodes impact the task performance. In this paper, we thus raise two questions: *Are background nodes necessary for target nodes classification? What roles do they play in the target classification task?*

To answer these questions, we conduct a comprehensive analysis using Graph Neural Network (GNN), one of the most popular graph models [41], for exploring the target-background issue. First of all, we observe a significant performance degradation when removing all background nodes, but a negligible impact of removing background nodes that are not neighboring the target nodes. Moreover, we find that the connection between target nodes plays a vital role in classification. Removing the background nodes that bridge between multiple target nodes results in a dramatic performance decline. Additionally, even background nodes neighboring a single target node would exhibit relatively higher feature correlation with

their corresponding target nodes and contribute to the classification. Detail experimental results and analyses can be seen in Section 2.

This exploration reveals two key insights: First, the majority of background nodes are redundant, while the nodes neighboring the target nodes are crucial for target classification. Second, background nodes contribute to the target nodes classification primarily in two ways: i) enhance the *connectivity* between targets as *bridging node*; ii) neighboring to single target node but have *feature correlation* with the target node as the *affiliation node* (illustrated in Figure 1 (c)). With this inspiration, we argue that it is possible to generate a small and highly informative subgraph (with original target nodes and far fewer background nodes) from original web graph. The generated graph contains rich information for target nodes classification and is also friendly for graph model deployment and storage. However, we still face two challenges. (1) Extracting subgraph from original one would inevitably cause semantic and structural information loss. How to properly fetch the subgraph with useful background nodes? (2) The fetched subgraph would also contain redundant structural and semantic information. How to condense the subgraph while preserving the essential information for target classification?

In this paper, we propose a novel *Graph-Skeleton* to generate a small, synthetic and highly-informative graph for target node classification. Specifically, following the intuition of empirical analysis, we formulate a principle for background node fetching, ensures that the extracted subgraph maintains both the structural connectivity and feature correlation via bridging and affiliation background nodes. After subgraph extraction, we propose three graph condensation strategies to condense the redundant structural and semantic information. The condensed synthetic graph (term as *skeleton*) contains sufficient information for target node classification and enjoys the benefits of small scale. Our main contributions are summarized as follows: (1) We first address a common challenge in real-world web applications: compressing the massive background nodes for classifying a small part of target nodes, to ease data storage, GNNs deployment and guarantee the performance. Empirical analysis explicitly indicates the contributions of background nodes to the target classification, i.e., enhancing target *structural connectivity* and *feature correlation* with target nodes, which provides a valuable guidance for background nodes fetching. (2) We propose a novel *Graph-Skeleton* for massive background nodes compression. It properly fetches the useful background nodes from massive web graph and performs background node condensation to eliminate information redundancy. (3) Extensive experiments on various web graphs demonstrate the effectiveness and efficiency of the proposed method. In particular, for MAG240M dataset with 0.24 billion nodes, our generated skeleton graph achieves highly comparable performance while only contain 1.8% nodes of the original graph.

## 2 EMPIRICAL ANALYSIS

In this section, we conduct empirical analyses to explore the target-background problem, for answering two key questions we raise above: *Are the background nodes necessary for target nodes prediction? What roles do they play in the target classification task?* We first analyze the overall contribution of background nodes in target classification, and then we explore what kind of background nodes are essential and how they contribute to the performance.

To ensure the generality of our analysis, we employ GNNs (GraphSAGE [16], GAT [33], GIN [36]) with three representative aggregation mechanisms, including mean, weight-based and summation, as the backbones for target nodes classification. The task is conducted on two datasets: (1) Financial loan network DGraph [21]. We follow the same task setting as the original dataset, i.e., fraudster identification among loan users, so that the users with loan action are regarded as target nodes (∼33%), while others are background nodes. (2) Academic citation network ogbn-arxiv [20]. We aim to predict the subject areas of papers published since 2018. In this case, papers published from 2018 are regarded as target nodes (∼46%), while papers before 2018 are background nodes. The detailed experimental settings are provided in Appendix A.3.

**Are Background Nodes Necessary for Target Classification?** We first evaluate the contribution of background nodes to the overall performance. Specifically, we delete all the background nodes by cutting background-to-background edges (B-B) and target-to-background (T-B) edges. In this way, the information propagation of each background node will be cut-off. As comparison, the random edge cut (cut ratio spans from 0 to 1) is implemented. As results depicted in Figure 1 (a), when cutting B-B (■), the performances of all GNNs show no significant decline and even presents slight improvement (DGraph) compared to the original graph, indicating the background nodes are indeed highly redundant and even contain noise. However, when cutting T-B (□), the performance presents a significant decline compared to the random edge cut (—). It reveals that the background nodes contain abundant information, which is essential to target node prediction.

**Background Nodes Contribute to Structural Connectivity Between Target Nodes.** For a comprehensive analysis, we additionally cut the target-to-target edges (T-T) to explore the dependency between the target nodes. One key observation is that the *structural connectivity* between target nodes plays an essential role in prediction. As shown in Figure 1 (a), the performance of cutting T-T (■) presents a significant decline compared to the original graph and random edge cut (—). Then, what role of the background nodes play in the target node classification? Inspired by the above observations, we cut the T-B edges where background nodes act as the 1-hop bridging nodes between two target nodes (i.e., T−✗−B−✗−T, BridB) to weaken the connectivity between targets. Consistent with T-T edge cutting, the performance of BridB cutting (■) also declines significantly (Figure 1 (a)), indicating that enhancing target connectivity via bridging background nodes contributes to task.

**Background Nodes Has Higher Feature Correlation with Neighboring Target Nodes.** From the experimental results in Figure 1 (a), we can still observe a performance gap between BridB cutting (■) and T-B edges cutting (□), i.e., the performance of BridB cutting outperforms that of cutting all background nodes. This indicates that apart from the background node bridging multiple targets, the background node neighboring to a single target node also contributes to the task.

From previous studies, numerous representative GNNs [14, 16, 27, 33] employ a repeated propagation process to integrate the feature information from neighboring nodes [39, 40]. This process promotes the similarity of features among neighboring nodes, leading to the creation of synthetic and robust node representations.

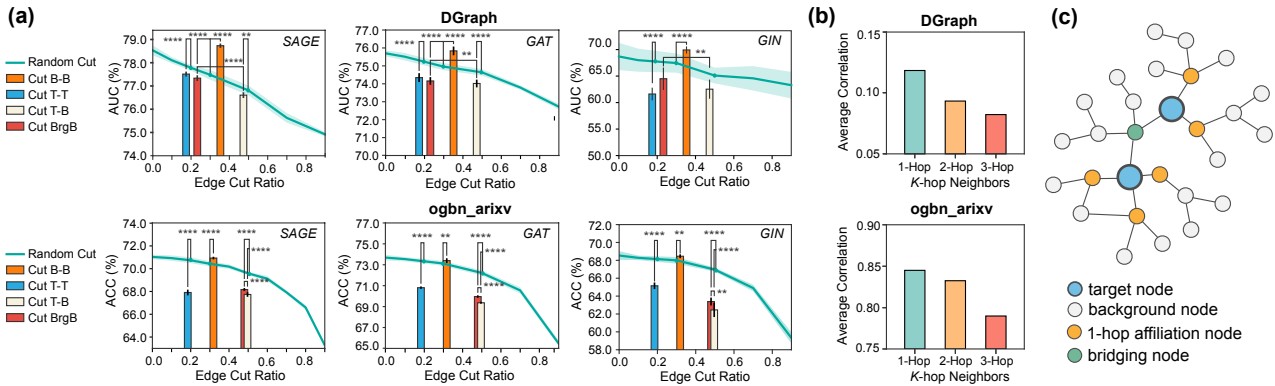

**Figure 1: (a) Explorations of background nodes influences. Upper: results on DGraph [21]. Lower: results on ogbn-arxiv [20]. (****$p$ < 1e-4, **$p$ < 1e-2, paired t-tests; errorbars represent the standard deviation). (b) Feature correlation between target nodes and their neighboring background nodes. (c) Illustration of target nodes and the corresponding essential background nodes.**

Following this, we hypothesize that the background nodes may exhibit higher *feature correlation* with their neighboring target nodes, enabling them to contribute during the propagation operation. To verify our hypothesis, we compute the Pearson correlation coefficient of features between target nodes and the background nodes with different neighboring hops. As the results shown in Figure 1 (b), the background nodes closer to the target nodes indeed have higher feature correlation, suggesting that the features of background nodes neighboring to a single target may correlated with feature of target itself, and thus contribute to the performance.

**Intuitions.** The analysis above provides us two key insights. First, the majority of background nodes are redundant, while the nodes neighboring to the target nodes are important to the target classification. Second, background nodes contribute to the target nodes classification primarily in two ways: i) enhance the *structural connectivity* between target nodes as *bridging node*; ii) neighbor to a solo target node with *feature correlation* to that target node as the *affiliation node* (illustrated in the Figure 1 (c)).

## 3 METHODOLOGY

**Problem Definition.** Given a large-scale graph $\mathcal{G} = (\mathcal{V}, \mathcal{E})$ and a specific node classification task, we can thus get a corresponding node set $\mathcal{T} = \{T_1, T_2, ..., T_n\}$, containing the nodes that are required to be classified, where $|\mathcal{T}| \ll |\mathcal{V}|$ in most of real-world scenarios. In this paper, we refer $\mathcal{T}$ as *target nodes*, and other nodes in $\mathcal{G}$ as background nodes $\mathcal{B} := \mathcal{V} \setminus \mathcal{T} = \{B_1, B_2, ..., B_{|\mathcal{V}|-n}\}$. The graph is also associated with node features $X \in \mathbb{R}^{|\mathcal{V}| \times d}$ and target node labels $Y \in \{0, ..., C-1\}^n$. Given that the majority of nodes are background nodes, our objective is to generate a synthetic graph $\mathcal{G}'$ that is highly informative while significantly reducing background nodes to alleviate the computational and storage burdens. This synthetic graph $\mathcal{G}'$ can be used to train graph models and classify the target nodes directly with comparable performance to the original graph $\mathcal{G}$. In this paper, we focus on the target nodes analysis in the real-world applications. Therefore we only compress the background nodes while preserving the whole original target nodes $\mathcal{T}$ in the generated synthetic graph $\mathcal{G}'$. This ensures that none of the target nodes are lost, which is crucial as it allows us

to trace and retain the specific information associated with each individual target node in the compressed synthetic graph $\mathcal{G}'$.

**Framework Overview.** To tackle the problem, we propose a novel *Graph-Skeleton* framework to generate a synthetic skeleton subgraph from massive web graph with much smaller size but rich information for target classification. The framework is illustrated in Figure 11. It first fetches all target nodes and a subset of background nodes to construct a vanilla subgraph (Figure 11, left). For proper background nodes fetching, we formulate a fetching principle following the inspirations of *structural connectivity* and *feature correlation* in Section 2. Then Graph-Skeleton condenses the graph information of the vanilla subgraph (Figure 11, middle) to reduce redundancy. Specifically, we design three graph condensation strategies (i.e., $\alpha, \beta, \gamma$) with condensation level ranging from low to high degree. The condensed graph (refer as *skeleton graph*) is highly informative and enjoys the benefits of small-scale for storage and graph model deployment (Figure 11, right).

### 3.1 Node Fetching

The observations in Section 2 reveal that the background nodes can be massive and highly redundant. To tackle this limitation, one natural idea is to reduce the graph size by fetching the essential background nodes and removing those that make little contribution to target classification. Inspired by the key observations of *structural connectivity* and *feature correlation*, we design a fetching principle to properly fetch the *bridging* and *affiliation* background nodes from massive original data as the first phase. Utilizing these nodes, we can construct a vanilla subgraph containing all target nodes and a small subset of background nodes. To alleviate the over-expansion issue, we customize the fetching depth and width with $d_1, d_2, K \in \mathbb{N}$. The fetching principle is formulated as follows:

> **Principle of What Background Node Will be Fetched:**
> **1. Structural connectivity:** *Bridges two or more target nodes within $d_1$-hop as bridging node.*
> **2. Feature correlation:** *K highest correlation background nodes neighboring to solo target node within $d_2$-hop as affiliation nodes.*

**Bridging Background Node Fetching.** Following principle.1, to fetch the bridging background nodes, our first step is to identify all *accessible background nodes* for each target node. Note that the

**Figure 2: Graph-Skeleton framework: it generates a synthetic skeleton subgraph from original graph with rich information for target prediction while enjoying the benefits of small scale. It first fetches the essential background nodes under the guidance of *structural connectivity* and *feature correlation* (Left), then condenses the information of background nodes (Middle). The generated skeleton graph is highly informative and friendly for storage and graph model deployment (Right).**

*accessible nodes* for a node $A$ refers to the nodes that can be reached from $A$ along a path only composed of background nodes. We will use this notion in the following paper. To this end, we utilize the breadth first search (BFS) [8] to find the shortest paths from each target node to its all accessible background nodes. By traversing all target nodes, the shortest paths set for each background node to its accessible target nodes can also be obtained. Let $B_j$ be one background node and is accessible to target nodes $T_{k_1}, T_{k_2}, ..., T_{k_i}$, with the corresponding shortest path set $PB_j = \{p_{k_1,j}, p_{k_2,j}, ..., p_{k_i,j}\}$. To verify whether $B_j$ aligns with Principle.1, we calculate the length of each path $d(PB_j) = \{d(p_{k_1,j}), ..., d(p_{k_i,j})\}$ ($d(\cdot)$ is the distance function), and sum up the minimum and second-minimum distance values of these paths as $sd_j = \min[d(PB_j)] + \min_{2nd}[d(PB_j)]$. If $sd_j \leq d_1$, it indicates that the background nodes $B_j$ bridges at least two target nodes within $d_1$-hop and will be regarded as the bridging node. By traversing all background nodes, a node subset $\mathcal{BR}$ containing all bridging background nodes can be obtained.

**Affiliation Background Node Fetching.** By conducting BFS for each target node $T_i$, we can obtain the shortest path set of $T_i$ containing paths to its all accessible background nodes. Let $J := \{j_1, j_2, ..., j_k\}$ be the indices of accessible background nodes to $T_i$, the corresponding shortest path set is $PT_i := \{p_{i,j}, j \in J\}$, where $p_{i,j}$ refers to the shortest path from $T_i$ to an accessible background node $B_j$. Following principle.2, we first pick the accessible background nodes with the shortest path distance within $d_2$, i.e., $\{B_m, m \in J, s.t., d(p_{i,m}) \leq d_2\}$. To fetch the most essential $K$ background nodes, we compute the feature Pearson correlation coefficient ($PCC$) for each picked $B_m$ with $T_i$, $PCC_{im} = \frac{cov(X[i]X[m])}{\sigma_{X[i]}\sigma_{X[m]}}$, where $X$ is the node feature matrix, $cov(\cdot)$ refers to the covariance and $\sigma$ refers to standard deviation. Then background nodes in $\{B_m\}$ with $K$ largest $PCC$ will be fetched as affiliation nodes of $T_i$. By traversing all target nodes, an affiliation background nodes subset $\mathcal{AF}$ can be obtained. Then we can construct a vanilla subgraph $\mathcal{G}' = (\mathcal{V}', \mathcal{E}')$ by preserving the target nodes $\mathcal{T}$ and the fetched background nodes $\mathcal{B}' = \{\mathcal{BR}, \mathcal{AF}\}$ within the original graph $\mathcal{G}$ (Figure 11 left, where $d_1$, $d_2$ and $K$ are set to 3, 1 and 2 respectively).

## 3.2 Graph Condensation

To reduce information redundancy, we develop a condensation process for the constructed vanilla subgraph $\mathcal{G}'$, which effectively condenses both structural and semantic information. Specifically, we propose three graph condensation strategies, denoted as $\alpha$, $\beta$, and $\gamma$, which provide varying degrees of condensation, ranging from low to high.

**Strategy-$\alpha$.** Following previous studies [5, 38], the number of equivalence classes can be utilized to measure the richness of information. Under this inspiration, we leverage the equivalence relationship of node pairs as a hint of information redundancy, enabling us to condense the semantic and structural information in the vanilla subgraph $\mathcal{G}'$. To do so, we first introduce the notion of node pair equivalence relation [5] on a graph $\mathcal{G} = (\mathcal{V}, \mathcal{E})$, with $X$ the node feature matrix.

*Definition 3.1 (Node Pair Equivalence Class).* Given a function family $\mathcal{F}$ on $\mathcal{G}$, define equivalence relation $\simeq_{\mathcal{F}}$ among all graph node pairs such that $\forall u, v \in \mathcal{V}, u \simeq_{\mathcal{F}} v$ iff $\forall f \in \mathcal{F}, f(\mathcal{G}, X) = f(\mathcal{G}, \tilde{X})$, where $\tilde{X} = X$ except $\tilde{X}[u] = X[v], \tilde{X}[v] = X[u]$.

Considering the equivalent node pairs share an identical structure within the graph, we argue that there is a large space for graph condensation. To this end, we propose a condensation strategy-$\alpha$, which leverages *multiple structure-set* ($MSS$) to captures the local structural information of each fetched background node in vanilla subgraph $\mathcal{G}'$. It allows us to identify background nodes with similar structural information and condense them into a synthetic node (shown in Figure 3). Specifically, for one background node $B_j \in \mathcal{B}'$ in $\mathcal{G}'$, we formulate the multiple structure-set $MSS_j$ via its accessible target nodes and the corresponding shortest path distances:

$$MSS_j = \{\langle T_i, d_{i,j}\rangle, ..., \langle T_k, d_{k,j}\rangle\}, \tag{1}$$

where $T_i, ..., T_k$ are the accessible target nodes of $B_j$, $d_{i,j}$ represents the shortest path distance between $B_j$ and $T_i$ (For simplicity, we use this notation in the following paper). For the background nodes with the same $MSS$, we claim that these background nodes belong to the same linear message path passing (LMPP) equivalence class. The detailed definition is given below.

*Definition 3.2 (Linear message passing operation).* Given two connected nodes $u, v$, define the linear message passing operation $f_{lmp}(u, v)$ from $u$ to $v$ as:

$$X'[v] \leftarrow f_{lmp}(X[v], X[u]) = \text{AGGREGATE}(\{X[v], X[u]\})W, \tag{2}$$

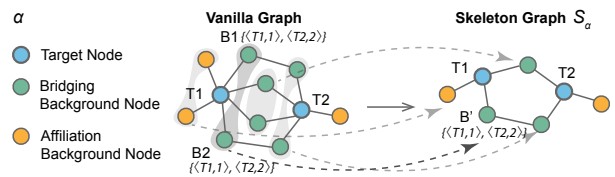

**Figure 3: Illustration of Strategy-$\alpha$, where the background nodes sharing a identical structural multiple-set ($MSS$) $\{\langle T, d \rangle\}$ (within a same shadow envelope) will be condensed into one synthetic node.**

where $W$ is a transformation matrix, AGGREGATE can be formulated as element-wise mean or summation pooling.

*Definition 3.3 (Linear message path passing).* Given a path $p = \langle u_0, u_1, ..., u_\ell \rangle, \forall u_i \in \mathcal{V}$, define the linear path passing functions $f_{spp}(X, p)$ aggregating node feature from $u_0$ to $u_\ell$ over $p$ as:

$$X'[u_i] \leftarrow f_{lmp}^i(X'[u_{i-1}], X[u_i])W^i, i \in 1, .., \ell. \quad (3)$$

*Definition 3.4 (LMPP equivalence class).* Given a family of linear message path passing functions $\mathcal{LMPP}$, two nodes $u, v \in \mathcal{V}$ and one common accessible node $K \in \mathcal{V}$, with the corresponding paths $p_{u,K} := \langle u, ..., K \rangle$ and $p_{v,K} := \langle v, ..., K \rangle$, let $P := p_{u,K}, p_{v,K}$, define LMPP equivalence relation $u \simeq_{\mathcal{LMPP},K} v$ iff $\forall f_{lmpp} \in \mathcal{LMPP}$, $f_{lmpp}(X, P) = f_{lmpp}(\tilde{X}, P)$ where $\tilde{X} = X$ except $\tilde{X}[u] = X[v], \tilde{X}[v] = X[u]$.

Here we relax the function family $\mathcal{F}$ in Definition 3.1 to linear message path passing functions $\mathcal{LMPP}$ over the paths for aggregation rather than the whole graph. Now we give a proposition to characterize the LMPP equivalence relation of background nodes.

PROPOSITION 3.5. *With $T \in \mathcal{T}$ denoting a target node in $G$, $\mathcal{B}$ denotes the set of background nodes, $\forall u, v \in \mathcal{B}$, if $MSS_u = MSS_v \neq \emptyset$, then $u \simeq_{\mathcal{LMPP},T} v$.*

We provide the proof of Proposition 3.5 in Appendix B. Proposition 3.5 and the proof suggests that for background nodes with same $MSS$, sharing one same path for linear message passing operation delivers the same aggregated features at target node as using their own original paths. In this case the original multiple paths for linear message passing are actually redundant.

Following this, we argue that the background nodes with the same $MSS$ may also leverage quite similar structure for information aggregation over the graph, and condensing this structural and feature information may also deliver similar aggregation results via nonlinear message passing operation. To this end, we condense the background nodes with the same $MSS$ into one synthetic node for reducing information redundancy. As an example shown in Figure 3, both $B_1, B_2$ have the identical multiple structure-set contents, i.e., $MSS_1 = MSS_2 = \{\langle T_1, 1 \rangle, \langle T_2, 2 \rangle\}$, and will be condensed into one synthetic node $B'$. To preserve the semantic information of condensed background nodes, we generate the synthetic node feature via aggregating the original features of the corresponding condensed background nodes. Let $\mathcal{B}'_k = \{B_i, ..., B_j\}$ be the set of background nodes with same $MSS$ to be condensed into a synthetic node $B'_k$, the feature of $x_{B'_k}$ is

$$x_{B'_k} \leftarrow \text{AGGREGATE}(\{x_v, \forall v \in \mathcal{B}'_k\}), \quad (4)$$

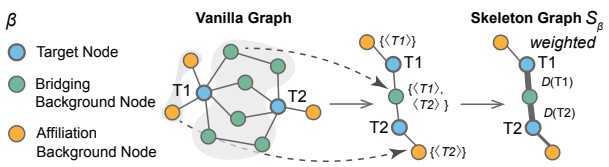

**Figure 4: Illustration of Strategy-$\beta$. The background nodes sharing the same structural multiple-set $MSS'$ $\{\langle T \rangle\}$ (within the same shadow envelope) will be condensed into one node. To maintain the relative distance information between different nodes, we encode the distance information of target nodes by weighting the edges of skeleton graph.**

where AGGREGATE($\cdot$) can be element-wise mean or summation pooling. By condensing all sets of $\mathcal{B}'$, the condensed skeleton graph $\mathcal{S}_\alpha$ can be obtained for storage and graph model deployment.

**Strategy-$\beta$.** While the strategy-$\alpha$ effectively reduces the graph redundancy, its compression effect is limited as only a portion of background nodes strictly share the same $MSS$ $\{\langle T, d \rangle\}$. To this end, we propose the second condensation strategy-$\beta$ with stronger condensation capacity over vanilla subgraph $\mathcal{G}'$. As a compromise, for each background node, we only involve its accessible target nodes while omitting the corresponding distance information in $MSS$, i.e., $MSS' = \{\langle T_i \rangle, ..., \langle T_j \rangle\}$. Similarly, for the background nodes with the same $MSS'$, they will be condensed into a synthetic node. As shown in Figure 4, the background nodes $B_1, B_2, B_3, B_4, B_5, B_6$ all have the same $MSS' = \{\langle T_1 \rangle, \langle T_2 \rangle\}$, which will be condensed into a synthetic node $B'$. The features generations of new condensed synthetic nodes follow the Eq 10.

However, it should be noted that in this strategy, the condensed graph would lose the relative distance information between nodes. For instance, the target nodes get closer when bridging background nodes are condensed together. Some target nodes that were originally several hops apart may be connected by a 1-hop bridging background node in the condensed graph.

To address this issue, we propose to encode the relative distance information between targets and backgrounds onto the edges in condensed graph. Let $\mathcal{B}'_k = \{B_i, ..., B_j\}$ be the set of background nodes in vanilla subgraph $\mathcal{G}'$ with the same $MSS'_k = \{\langle \mathcal{T}_k \rangle\}$, which will be condensed into a synthetic node $B'_k$. Given one accessible target node $T_m \in \mathcal{T}_k$ connected with $B'_k$ via edge $e'_{m,k}$, and the shortest distance set of $\mathcal{B}'_k$ to $T_m$ is $D_m := \{d_{m,i}, ..., d_{m,j}\}$. Then the weight of edge $e'_{m,k}$ is formulated as: $w'_{m,k} = \sum \frac{1}{d}, \forall d \in D_k$, which can be utilized to weight the features during aggregation in downstream graph models. To guarantee the scale consistency, normalization based on rows and columns is also required for the edge weights. After condensation for all sets of $\mathcal{B}'$, we can obtain the condensed skeleton graph $\mathcal{S}_\beta$.

**Strategy-$\gamma$.** Although the condensation-$\beta$ strategy effectively reduces the size of bridging background nodes, there is still room for further condensation of affiliation background nodes. Numerous representative GNNs employ the propagation process to merge the information from neighbors [16, 27]. Intuitively, aggregating the features of affiliation nodes onto the neighboring target node directly would deliver a similar result with the recursive aggregation.

**Table 1: Target node classification comparisons on the original and the compressed graphs with fixed *BCR*s. "OOM": out of memory. DGraph uses AUC (%) for evaluation, other datasets use ACC (%).**

| Datasets | GNN | Original | Skeleton-γ | Random | Cent-P | Cent-D | Schur-C | GC-VN | GC-AJC | GCond | DosCond | GPA |
|---|---|---|---|---|---|---|---|---|---|---|---|---|
| DGraph BCR=15% | SAGE [16] | 78.7±0.1 | **78.2±0.1** | 74.2±0.1 | 74.6±0.1 | 74.5 ±0.1 | 75.4±0.1 | OOM | OOM | OOM | OOM | OOM |
| | GCN [27] | 73.4±0.2 | **74.4±0.1** | 72.1±0.1 | 72.6±0.1 | 72.5±0.1 | 72.8±0.2 | OOM | OOM | OOM | OOM | OOM |
| | GAT [33] | 75.9±0.4 | **76.6±0.6** | 73.3±0.3 | 73.9±0.4 | 73.8±0.4 | 74.2±0.5 | OOM | OOM | OOM | OOM | OOM |
| | NAGphormer [3] | 76.5±0.1 | **77.2±0.1** | 73.7±0.1 | 74.49±0.1 | 74.21±0.1 | 75.1±0.2 | OOM | OOM | OOM | OOM | OOM |
| ogbn_arxiv BCR=7% | SAGE [16] | 71.0±0.5 | 68.8±0.4 | 66.7±0.2 | 68.0±0.3 | 67.9±0.3 | **68.9±0.4** | 67.5±0.5 | 68.3±0.4 | 64.2±0.3 | 61.2±0.3 | OOM |
| | GCN [27] | 71.3±0.2 | **69.2±0.3** | 67.8±0.3 | 68.1±0.2 | 69.0±0.3 | 68.0±0.3 | 68.1±0.2 | 68.0±0.2 | 64.3±0.4 | 61.5±0.5 | OOM |
| | GAT [33] | 72.1±0.1 | **70.9±0.1** | 70.1±0.1 | 70.3±0.2 | 70.2±0.1 | 70.5±0.2 | 70.1±0.2 | 70.3±0.2 | 62.4±0.5 | 59.3±0.4 | OOM |
| | SIGN [10] | 71.8±0.1 | **69.3±0.2** | 68.8±0.2 | 69.0±0.1 | 69.2±0.2 | 69.1±0.2 | 69.1±0.1 | 69.2±0.2 | 63.6±0.3 | 60.4±0.3 | OOM |
| DBLP BCR=12% | SAGE [16] | 84.1±0.4 | **82.0±0.3** | 79.3±0.2 | 80.0±0.4 | 79.8±0.5 | 78.3±0.4 | 77.9±0.8 | 79.8 ± 0.5 | 78.3±0.7 | 76.1±0.5 | 80.2±0.4 |
| | GCN [27] | 81.7±0.3 | **82.1±0.3** | 79.5±0.3 | 80.1±0.2 | 80.0±0.3 | 67.1±0.8 | 81.2±0.7 | 81.4±0.5 | 79.5±0.9 | 76.5±0.6 | 79.7±0.3 |
| | GAT [33] | 79.5±0.3 | **79.3±0.3** | 79.0±0.5 | 78.9±0.4 | 79.1±0.4 | 57.1±1.6 | 71.2±0.6 | 67.9±1.2 | 74.5±1.0 | 74.2±0.8 | 78.5±0.5 |
| IMDB BCR=36% | SAGE [16] | 55.2±0.6 | 55.6±1.2 | 48.8±0.8 | 51.1±0.8 | 51.3±0.8 | 53.1±0.7 | 55.4±0.5 | **56.6±0.3** | 49.1±0.8 | 53.2±1.0 | 51.9±0.9 |
| | GCN [27] | 56.9±0.7 | **56.9±1.2** | 49.8±0.8 | 51.6±0.7 | 51.8±0.7 | 53.4±0.8 | 55.5±0.5 | 56.4±0.5 | 50.6±0.6 | 53.5±0.8 | 52.5±0.7 |
| | GAT [33] | 57.2±0.6 | **54.5±1.0** | 48.2±0.7 | 50.7±0.8 | 51.1±0.7 | 51.6±1.8 | 52.3±0.6 | 52.8±0.6 | 48.9±0.7 | 50.2±0.6 | 49.4±0.8 |
| ogbn_mag BCR=40% | R-GCN [32] | 46.0±0.7 | **46.2±0.4** | 22.4±0.9 | 24.8±0.8 | 25.1±0.7 | 39.5±0.4 | OOM | OOM | OOM | OOM | OOM |
| | GraphSaint [37] | 43.2±0.5 | **43.9±0.3** | 9.1±1.1 | 13.6±1.5 | 11.2±1.2 | 36.7±0.5 | OOM | OOM | OOM | OOM | OOM |
| | Cluster-GCN [7] | 38.5±0.2 | **39.5±0.2** | 35.2±0.3 | 36.0±0.2 | 35.9±0.2 | 33.5±0.3 | OOM | OOM | OOM | OOM | OOM |
| MAG240M BCR=1% | R-GAT [32] | 70.0 | **68.5** | 59.0 | 59.8 | 60.1 | OOM | OOM | OOM | OOM | OOM | OOM |
| | R-SAGE [32] | 69.4 | **68.2** | 59.4 | 59.6 | 60.4 | OOM | OOM | OOM | OOM | OOM | OOM |

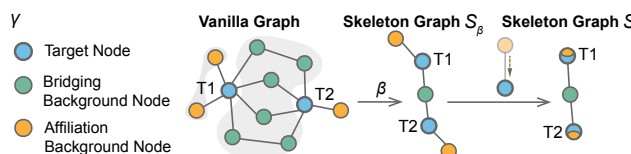

**Figure 5: Illustration of Strategy-γ. Based on condensation-β, we further condense the affiliation nodes to the corresponding target node.**

Under this inspiration, we propose the third strategy-γ for condensation. Given the vanilla subgraph $\mathcal{G}'$, we first perform strategy-β to condense the bridging background nodes, then we condense the affiliation nodes in the generated skeleton graph $\mathcal{S}_\beta$. Specifically, given one target node $T_k$ in $\mathcal{S}_\beta$, with its corresponding affiliation background node set $\mathcal{AF}_k$, we update the feature of target $T_k$ by incorporate its feature with the features of $\mathcal{AF}_k$, and then remove $\mathcal{AF}_k$ in $\mathcal{G}'$ (shown in Figure 5), to eliminate the massive affiliation nodes while maintaining most of the original correlation information from affiliation nodes. The condensed feature of $T_k$ via aggregating the original features with $\mathcal{AF}_k$ is formulated below:

$$x_{T_k} \leftarrow \text{AGGREGATE}(\{x_{T_k} \cup \{x_u, \forall u \in \mathcal{AF}_k\}\}), \quad (5)$$

AGGREGATE($\cdot$) can be element-wise mean or summation pooling. Finlay, we can obtain the condensed skeleton graph $\mathcal{S}_\gamma$, which is highly informative and friendly to storage and model deployment. We provide the time complexity analysis in Appendix C.

## 4 EXPERIMENTS

**Experimental protocol.** To comprehensively evaluate the performance of the proposed *Graph-Skeleton*, we conduct the target nodes classification task on six web datasets: DGraph [21], ogbn-mag [20], ogbn-arxiv [20], MAG240M [19], DBLP [11] and IMDb [11], spanning across multiple domains. Based on the downstream task, we

**Table 2: Statistics of datasets.**

| Dataset | Nodes | Edges | Target Definition | Targets |
|---|---|---|---|---|
| DGraph | 3,700,550 | 4,300,999 | Loan Users | 1,225,601 |
| ogbn-arxiv | 169,343 | 1,166,243 | Papers (since 2018) | 78,402 |
| IMDB | 11,616 | 17,106 | Movies | 4,278 |
| DBLP | 26,108 | 119,783 | Authors | 4,057 |
| ogbn-mag | 1,939,743 | 21,111,007 | Papers | 736,389 |
| MAG240M | 244,160,499 | 1,728,364,232 | arxiv papers | 1,398,159 |

can obtain the corresponding target nodes (required to be classified) and background nodes. The basic information of datasets and how we select the target nodes are listed in Table 2. We also provide a more detailed description in Appendix A.1. Note that in our study, we only compress the background nodes, and all target nodes are preserved in the generated skeleton subgraph. In this case, the generated skeleton and original graph contain the same target nodes for classification.

**Experimental Setup.** We compare the downstream target node classification performance with original graph and other graph compression baselines including coreset methods (*Random*, Centrality Ranking [13] with PageRank centrality (*Central-P*) and degree centrality (*Central-D*)), graph coarsening methods (variation neighborhoods coarsening (*GC-VN*) [22], Algebraic JC coarsening (*GC-AJC*) [22], spectral coarsening with Schur complement (*Schur-C*) [44], graph condensation methods (*GCond* [25], *DosCond* [24]) and graph active learning method (*GPA* [18]). Note that our goal is to compress the background node compression while maintain all target nodes. The compression rate is indicated by *background node compression rate* (*BCR*) (ratio of synthetic background nodes to original background nodes, details in A.2). For a fair comparison, we keep the *BCR* same across all methods. More details of the baselines and settings can be found in Appendix A.4.2.

After obtaining the compressed graphs by above methods, we adopt the GNNs to evaluate their target classification performance. Considering different datasets would be applicable to different GNNs, for DGraph, ogbn-axiv, ogbn-mag and MAG240M, we select

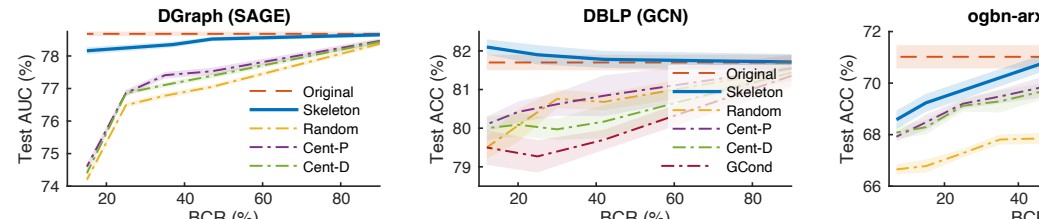

Figure 6: Comparisons of graph compression methods with varied *BCR*s for GNNs inference.

Table 3: Memory costs for data storage.

| DGraph | Target Feat | Background Feat | Adj Matrix | AUC (SAGE) |
|---|---|---|---|---|
| Original | 166.3 MB | 336.6 MB | 128.8 MB | 78.7±0.1 |
| Skeleton-$\gamma$ (BCR: 15%) | | 50.1 MB | 52.4 MB | 78.2±0.1 |
| **ogbn-arixv** | Target Feat | Background Feat | Adj Matrix | ACC (SAGE) |
| Original | 40.1 MB | 46.5 MB | 37.0 MB | 71.0±0.5 |
| Skeleton-$\gamma$ (BCR: 7%) | | 3.2 MB | 12.3 MB | 68.6±0.4 |
| **DBLP** | Target Feat | Background Feat | Adj Matrix | ACC (GCN) |
| Original | 1.0 MB | 5.6 MB | 3.8 MB | 81.7±0.3 |
| Skeleton-$\gamma$ (BCR: 12%) | | 0.7 MB | 0.2 MB | 82.1±0.3 |
| **IMDB** | Target Feat | Background Feat | Adj Matrix | ACC (GCN) |
| Original | 52.4 MB | 89.9 MB | 0.6 MB | 56.9±0.7 |
| Skeleton-$\gamma$ (BCR: 36%) | | 34.4 MB | 0.4 MB | 56.9±1.2 |
| **ogbn-mag** | Target Feat | Background Feat | Adj Matrix | ACC (GraphSaint) |
| Original | 377.0 MB | 616.1 MB | 674.9 MB | 43.2±0.5 |
| Skeleton-$\gamma$ (BCR: 40%) | | 245.6 MB | 505.8 MB | 43.9±0.3 |
| **MAG24M** | Target Feat | Background Feat | Adj Matrix | ACC (R-SAGE) |
| Original | 2.05 GB | 372.97 GB | 55.95 GB MB | 69.4 |
| Skeleton-$\gamma$ (BCR: 1%) | | 4.65 GB | 2.69 GB | 68.2 |

the base GNNs on their respective official leaderboards for evaluation. For IMDB and DBLP, we adopt the most representative GNNs (GCN [27], GraphSAGE [16], GAT [33]) for evaluation. The target classification performance of DGraph is evaluated by AUC (%), and other datasets are evaluated by ACC (%).

## 4.1    Graph Compression Comparison

We first report target node classification results of compressed graphs under fixed compression rate (*BCR*) on six datasets in Table 1, where we compare the performance of Graph-Skeleton using condensation strategy-$\gamma$ (Skeleton-$\gamma$ in short, which has highest compression rate) to other baselines. As we can see, Skeleton-$\gamma$ presents strong ability of scaling up GNNs to all datasets, including large-scale graph MAG240M with ~0.24 billion nodes. It also achieves superior target classification performance compared to other compression baselines under similar *BCR*. Besides, compared to other graph coarsening and compression methods being significantly hindered by heavy memory and computational loads, our method is more friendly for deployment on large-scale web graphs. Moreover, compared to the original graph, Skeleton-$\gamma$ also presents highly comparable or even better target classification performance with a notably smaller number of background nodes.

Additionally, we report the target classification results of compressed graph with varied *BCR*s in Figure 6. By selecting different condensation strategies of Graph-Skeleton with different fetching depths ($d1, d2$), we can flexibly achieve different compression rates (details in Appendix A.4.5). It can be easily observed that our method significantly outperforms other methods in a wider range of compression rate.

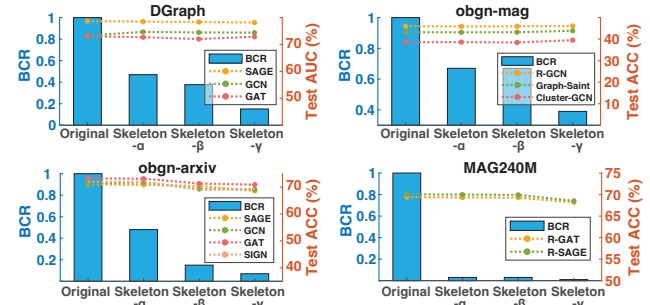

Figure 7: Compression performance of Graph-Skeleton with three condensation strategies: $\alpha, \beta, \gamma$. The bars indicate the background nodes compression rate (BCR), dashed lines with points indicate target node classification performance.

## 4.2    Storage Costs

We report the memory costs for web graph storage of original graph and Skeleton-$\gamma$ under the fixed *BCR* on six datasets. To present the results more intuitively, we decouple the storage costs of graph data into three main aspects: costs of target nodes features, background nodes features and graph adjacency matrix. The results are shown in Table 3. As we can see, Skeleton-$\gamma$ significantly reduces the memory cost of background nodes features and graph adjacent matrix (green in Table 3). Since we preserve all the target node in compressed Skeleton-$\gamma$, the storage cost of target nodes feature keeps the same with the original graph. On the other hand, Skeleton-$\gamma$ also achieves close or even better performance compared to the original data. This highlights the effectiveness of our proposed method in preserving the essential information for target node classification.

## 4.3    Studies on Three Condensation Strategies

In this section, we investigate the compression performance of three proposed condensation strategies $\alpha, \beta, \gamma$ of Graph-Skeleton. Specifically, we use the same vanilla subgraph as input and use three strategies for condensation. The results on four datasets are depicted in Figure 7, where the left axis (blue) presents the background nodes compression rate (*BCR*, bar) and right axis (red) presents the target node classification performance (dashed lines). As we can see, Skeleton-$\alpha, \beta, \gamma$ all present highly competitive target node classification performance with the original data, indicating the effectiveness of three proposed strategies for condensation. Generally the Skeleton-$\alpha$ presents the best classification performance within three strategies due to fewer information losses. On the other hand, Skeleton-$\gamma$ also well approximates the original performance under all tested downstream GNNs while with a notably higher *BCR*, showing aggregating the features of affiliation background nodes

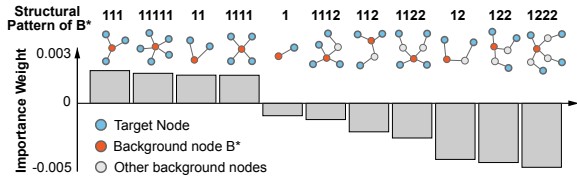

**Figure 8: The importance weights of top 10 structural patterns of background nodes in $\mathcal{S}_\alpha$ of DGraph. The corresponding structural patterns are visualized upper the bars.**

onto the neighboring target node can well preserve the original background node information.

## 4.4 What Kinds of Background Nodes Are Essential in Real-World Cases?

Due to different properties of the web graph data, the condensed skeleton graph structure also vary dramatically from each other. Since none of the target nodes are lost in our compressed skeleton graph, it allows us to trace and investigate some important information and patterns relevant to the target classification upon the generated skeleton graphs. In this section, we utilize the distance information in each background node' multiple structure-set $MSS$ in skeleton graph $\mathcal{S}_\alpha$ to represent the target-background structural patterns and leverage the attention mechanism to explore the importance of each structural pattern to the task. Given background node $B$ with $MSS = \{\langle T_i, d_i \rangle, \langle T_j, d_j \rangle\}$, define its structural patterns as $\{d_i, d_j\}$, indicating it has two accessible target nodes with shortest distances of $d_i, d_j$ respectively. For each structural pattern class, we compute its importance weights by averaging the attention weight of background nodes with the corresponding pattern class in GNN. The implementation details are provided in A.6.

**Exploration on DGraph.** We take DGraph [21] as a case study to investigate how the fetched background nodes contribute to the classification. Figure 8 presents the importance weights of different structural patterns maintained by the background nodes. For the top four structural patterns, i.e., {1,1,1}, {1,1,1,1,1}, {1,1} and {1,1,1,1}, the background nodes all act as the 2-hop bridging nodes, revealing the importance of connectivity between target nodes. This observation is also consistent with the exploration results in Section 1. Moreover, the attention scores of these four patterns are quite close, suggesting that the 2-hop bridging nodes with different degrees might play similar roles in the task. These observations can offer a good explanation for the classification task in the financial scenario where the social relations between users are crucial for fraud detection. Concretely, for one node connected with fraud users, the likelihood that it and its direct neighbors are fraudsters will increase significantly. Moreover, if most neighbors of one user are frauds, it is likely to be a fraud intermediary agency. Another case study of paper citation network is provided in Appendix E.4.

## 5 RELATED WORKS

In many web graph mining scenarios, web graph data is massive, and only a small ration of nodes are actually need to be analyzed. In this paper, we argue that a small synthetic subgraph would also be sufficient to trace, retain the information of each target node for classification. As far as we are aware, this is the first work exploring

the contributions of background nodes to the target classification, and compressing the graph while preserving essential information for each original target node. In the following parts, we will briefly introduce the mainstream methods of graph size reduction and discuss their differences from our work.

**Coreset Selection.** Up to the present, various coreset selection methods, including k-Means [17], centrality rank [1, 13], random selection are designed to select a subset of essential samples to reduce the data size. However, these methods either ignore the web graph structural information (k-Means, random selection) or ignore the node semantic (feature) information (centrality rank), leading to unsatisfied selection results on graph data. Moreover, background nodes contribute to the target classifications in different ways (e.g., structural connectivity enhancement, feature correlation), while the above methods ignore this vital knowledge.

**Graph Coarsening.** Graph coarsening reduces the number of graph nodes while preserving some important properties in the original graph. The theoretical approximation guarantees on spectrum or structure have been studies in some previous coarsening studies [2, 26, 29, 44]. Specifically, Zhu et al. propose a sparsifier which leverages schur complement construction to approximate the shortest distances between each pair of vertices in terminal set [44]. Nevertheless, these graph reduction methods only focus on the approximation of structure while ignoring the node features, which are not tailored to the node classification tasks in web mining. Huang et al. proposes a coarsening model for semi-supervised node classification, which merges the original nodes into super-nodes along with averaged node features for graph reduction [22].

**Graph Condensation.** Recently, graph condensation has also been studied. Jin et al. develop a graph condensation method based on gradient matching to imitate the GNNs training trajectory on large graph [25], and further extend the method to one-step gradient matching [24]. Nevertheless, these methods would inevitably lose original target nodes during reduction. For coarsening model [22], it is prone to merge the target nodes into one super-nodes. For the condensation methods [24, 25], they can only compress the target nodes since only the target nodes' labels are effective for gradient matching. However, in our study we aim to shrink the size of background nodes while maintaining all target nodes. Besides, these methods require large memory cost with higher time complexity during graph reduction, yet our method is efficient and effective, which is much easier to implement on very large graphs.

## 6 CONCLUSION

In this paper, We focus on a common challenge in web graph mining applications: compressing the massive background nodes for a small part of target nodes classification, to tackle the storage and model deployment difficulties on very large graphs. Empirical analysis explicitly reveals the contributions of critical background nodes to the target classification, i.e., enhancing target *structural connectivity* and *feature correlation* with target nodes. With these inspirations, we propose a novel *Graph-Skeleton*, which properly fetches and condenses the informative background nodes, so that the generated graph is small-scale but sufficient to trace, retain the information of each target node for classification. Extensive experiments well indicates the effectiveness of our proposed method.

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

# A  DETAILED EXPERIMENTAL SETTINGS

## A.1  Datasets

We evaluate the proposed framework on four datasets, i.e., DGraph[1], ogbn-mag, ogbn-arxiv, IMDB[2], DBLP[3] and MAG240M. Since all the datasets have no explicit definitions of the target & background, we manually define the target & background settings according to the usage scenarios of datasets. The datasets and their settings are detailed below:

(1) DGraph [21]: A large-scale financial graph dataset, with 3,700,550 nodes and 4,300,999 directed edges. There are four classes of nodes: Class 0 are normal loan users and Class 1 are loan fraud users, the other two classes of nodes are non loan users. In our experiment, we follow the task in the original dataset, i.e., identify the frauds among the users with loan record. In this case, the loan users are the target nodes required to be classified (~33% of all nodes), while the other users are set as the background nodes. The target nodes are split randomly by 7:1.5:1.5 for training, validation and testing, which is identical with the original dataset.

(2) ogbn-arxiv [20]: The citation arxiv network contains 169,343 nodes and 1,166,243 edges. Each node is an arXiv paper and each directed edge indicates that one paper cites another one. In this study, we define the task as predicting the 40 subject areas (e.g., cs.AI, cs.LG, and cs.OS) of each target paper published since 2018. Thus papers published since 2018 (~46%) are the target nodes, and papers published before 2018 are background nodes. We keep the test set identical to the original dataset (i.e., paper published since 2019), and randomly split the remaining nodes by 8:2 for training and validation.

(3) IMDB [11]: IMDB is an online database about movies and television programs, including information such as cast, production crew, and plot summaries. We follow the data usage in [11], which adopts a subset of IMDB scraped from online, containing 4278 movies, 2081 directors, and 5257 actors. Movies are labeled as one of three classes (Action, Comedy, and Drama). In our study, we set the movie nodes as the target nodes while other nodes are regarded as background nodes. For semi-supervised learning, we follow the split in [11]: movie nodes are divided into training, validation, and testing sets of 400 (9.35%), 400 (9.35%), and 3478 (81.30%) nodes, respectively.

(4) DBLP [11]: A computer science bibliography website. We follow the data usage in [11], which adopts a subset of DBLP extracted by [12, 23], containing 4057 authors, 14328 papers, 7723 terms, and 20 publication venues. In our study, we set the author nodes as the target nodes while other nodes are regarded as background nodes. The authors are divided into four research areas (Database, Data Mining, Artificial Intelligence, and Information Retrieval). For semi-supervised learning models, we follow the split in [11]: author nodes are divided into training, validation, and testing sets of 400 (9.86%), 400 (9.86%), and 3257 (80.28%) nodes, respectively.

[1]Dgraph: https://dgraph.xinye.com/introduction
[2]IMDB: https://www.imdb.com/
[3]DBLP: https://dblp.uni-trier.de/

(5) ogbn-mag [20]: A heterogeneous academic graph contains 1,939,743 nodes and 21,111,007 edges, with four types of nodes (entities) -papers, authors, institutions and fields of study. In our study, we aim to predict the venue of each paper. Hence, the paper nodes are regarded as target nodes (~ 38 %), and the other nodes (authors, institutions and fields), on the contrary, are background nodes for information supplement. We follow the split settings in the original dataset, i.e., training models to predict venue labels of all papers published before 2018, validating and testing the models on papers published in 2018 and since 2019, respectively.

(6) MAG240M [19]: A massive scale heterogeneous map with 240 million nodes and 1.7 billion edges. Nodes are composed of papers, authors and institutions in an academic graph. In our study, we aim to predict arxiv papers' subject areas. Thus the arxiv paper nodes (0.6%, 1.4 million) are considered as target nodes. We follow the original dataset split that the papers published earlier than 2018 are training sets, while articles published in 2018 and since 2019 are validation sets and test sets, respectively.

## A.2  Evaluation Metrics

We evaluate the proposed performance based on several metrics. For target classification tasks, the performances of ogbn-mag, ogbn-arxiv and MAG240M [20] are evaluated by accuracy metric (ACC). Due to the imbalance of the positive and negative samples in Dgraph, it uses the area under curve (AUC) metric for performance evaluation.

To access the graph compression performance, we utilize the background node compression rate ($BCR$) to indicate the compression effectiveness, which is defined as:

$$BCR = \frac{|\mathcal{B}_S|}{|\mathcal{B}_O|} \tag{6}$$

where $\mathcal{B}_S$ and $\mathcal{B}_O$ denotes the background node sets in generated skeleton graph and original graph. Please note that our goal is to maintain the whole target nodes in the generated synthetic graph, so we just focus on the background nodes compression.

## A.3  Experimental Settings in Section 2

Recently Graph Neural Networks (GNNs) [41] have become one of the most popular and powerful methods for data mining and knowledge discovery on graph data. Therefore we conduct our exploration of target-background issue in Section 2 based on the GNN models.

Concretely, we choose including GraphSAGE [16], GAT [33] and GIN [36], which utilized three representative aggregation mechanisms (mean, weight-based, summation), as the backbone model, and deploy the model on two datasets for target nodes classification:

- Financial loan network DGraph, where we follow the task setting of the original dataset [21], i.e., fraudster identification among loan users. In this case the users with loan action are regarded as target nodes (~33%), while others are viewed as background nodes.
- Academic citation network ogbn-arxiv [20], where we aim to predict the subject areas of papers published since 2018.

**Table 4: Hyperparameters of GNNs on Dgraph**

| # Params | SAGE | GAT | GIN |
|---|---|---|---|
| layers | 3 | 3 | 2 |
| hidden dim | 256 | 128 | 64 |
| head num | - | 3 | - |
| sample size | [10, 10, 10] | [10, 10, 10] | [10, 10] |
| batch size | 65,536 | 16,384 | 100,000 |
| epochs | 200 | 100 | 100 |
| learning rate | 0.01 | 0.01 | 0.02 |
| dropout | 0.0 | 0.5 | 0.5 |

**Table 5: Hyperparameters of GNNs on ogbn-arxiv**

| # Params | SAGE | GAT | GIN |
|---|---|---|---|
| layers | 3 | 3 | 2 |
| hidden dim | 256 | 256 | 256 |
| head num | - | 3 | - |
| sample size | full | full | full |
| batch size | full | full | full |
| epochs | 500 | 100 | 1500 |
| learning rate | 0.005 | 0.002 | 0.005 |
| dropout | 0.5 | 0.75 | 0.5 |

In this case, papers published from 2018 are regarded as target nodes (~46%), while papers before 2018 as the background nodes.

To analyze the contributions of background nodes to target nodes, we cut the edges between different types of nodes and deploy the cutted graphs on the GraphSAGE model. Specifically, we cut the edges within the original graph including: (1) Random Cut: randomly drop the edges within the graph with the edge drop rate spanning from 0 to 1. (2) B-B Cut: drop the edges between background nodes. (3) T-B Cut: drop the edges between target and background nodes. (4) BridB Cut: drop the edges where background nodes act as the 1-hop bridging nodes between two target nodes. The model hyperparameter settings of two datasets are listed in Table 4 and Table 5.

## A.4 Experimental Settings in Section 4

*A.4.1 Graph-Skeleton Settings in Fixed BCR Experiments.* For the evaluation of proposed Gpaph-Skeleton, we first compare the performance of baselines in a fixed *BCR* manner in Section 4.1, Table 1. Specifically, we choose the Gpaph-Skeleton using condensation strategy-$\gamma$ (Skeleton-$\gamma$ in short, with highest compression rate) to compare with other baselines. During node fetching phase (Section 3.1), we set the fetching depth and width as $d_1 = 2, d_2 = 1, K = 5$ to generate the vanilla subgraph $\mathcal{G}'$. For $\gamma$ condensation, we choose mean operation as aggregation function in Equation 5. The statistics of the final synthetic skeleton graph $\mathcal{S}_\gamma$ are shown in Table 6. For baselines of Random, Centrality Ranking and GPA, we control the query budgets of background node selection to keep the *BCR* being same with that of skeleton graph $\mathcal{S}_\gamma$. For Graph Coarsening, GCond and DosCond, we control the compression rate to keep the synthetic graphs having the same total size (# Total after compression) with skeleton graph $\mathcal{S}_\gamma$. These synthetic graphs are considered to have the same *BCR* for comparison.

*A.4.2 Baselines and Compression Settings.* To compare the compression performance of our method, we choose the following baselines for graph compression:

- Random: Randomly selects the background nodes from original data, where all target nodes can be preserved in compressed graph. The background nodes compression rate (*BCR*) can be easily controlled by selection number.
- Centrality Ranking [13]: Selects the background nodes according to the centrality ranking scores. Specifically, PageRank centrality (Central-P) and degree centrality (Central-D) are adopted for centrality ranking. The *BCR* can be easily controlled by selection number.

- Schur Complement [44]: A communication-efficient distributed algorithm for constructing spectral vertex sparsifiers (Schur complements), which closely preserve effective resistance distances on a subset of vertices of interest in the original graphs. To reduce the computational complexity, we employ the rLap [28] for schur complement approximation. The *BCR* can be controlled by controlling the size of Schur complements.
- Graph Coarsening [22]: Merges the massive nodes into fewer super-nodes to reduce the graph size. Specifically, we implement the variation neighborhoods coarsening (GC-VN) and Algebraic JC coarsening (GC-AJC) [30] in the original paper respectively. Since the graph coarsening is prone to delete the target nodes by merging them into the super-nodes during compression, we keep the total size of compressed graph to be same with that of skeleton during comparison while not strictly control the *BCR* to be the same. But we still regard them having the save *BCR* when reporting results. Since the target nodes are different after compression, we only utilize the synthetic graph for downstream model training and relax the model inference based on the original graph to obtain the results of each original target node.
- GCond [25]: Condenses the graph based on multi-step gradient matching for imitating the GNNs training trajectory on the original graph. This method generates the features and adjacent matrix of synthetic graph by gradient matching. Since the gradient matching task is implemented based on the labels of target nodes, this method can only compress the target nodes. Similarly, we keep the total size of compressed graph to be same with that of skeleton while not strictly control the *BCR* to be the same for comparison. But we still report them to have same *BCR* in the results. To have the results of all target nodes, we only utilize the synthetic graph for model training while relaxing the model inference based on the original graph.
- DosCond [24]: Condenses the graph based on one-step gradient matching for generating the features and adjacent matrix of synthetic graph. Similarly, this method can only compress the target nodes. and we keep the total size of compressed graph to be same for comparison. But we still report them to have same *BCR* in the results. To have the results of all target nodes, we utilize the synthetic graph for model training while using the original graph for inference.
- GPA[18]: A graph active learning framework selecting nodes for labelling. Here we use it for background node selection.

**Table 6: Statistics of the synthetic skeleton graphs in fixed *BCR* experiments**

|  | DGraph | ogbn-arxiv | DBLP | IMDB | ogbn-mag | MAG240M |
|---|---|---|---|---|---|---|
| # Target | 1,225,601 | 78,402 | 4,057 | 4,278 | 736,389 | 1,398,159 |
| # Original Background | 2,474,949 | 90,941 | 22,071 | 7,338 | 1,203,354 | 242,762,340 |
| # Compressed Background | 373,015 | 6,349 | 2,690 | 2,810 | 479,861 | 2,970,934 |
| # Total after compression | 1,598,616 | 84,751 | 6,747 | 7,088 | 1,216,250 | 4,369,093 |
| *BCR* | 0.150 | 0.069 | 0.121 | 0.362 | 0.398 | 0.012 |

**Table 7: Hyperparameters for GNNs (a)**

|  | DGraph [21] | | | ogbn-arxiv [20] | | | | MAG240M [19] | |
|---|---|---|---|---|---|---|---|---|---|
|  | **SAGE** | **GCN** | **GAT** | **SAGE** | **GCN** | **GAT** | **SIGN** | **R-GAT** | **R-SAGE** |
| **layers** | 3 | 3 | 3 | 2 | 2 | 3 | 2 | 2 | 2 |
| **hidden dim** | 256 | 64 | 128 | 256 | 256 | 250 | 512 | 1024 | 1024 |
| **head num** | - | - | [4,2,1] | - | - | 3 | - | 4 | - |
| **sample size** | [10,10,10] | Full | [10,10,10] | Full | Full | [10,10,10] | Full | [25,15] | [25,15] |
| **batch size** | 65,536 | Full | 16,384 | Full | Full | 512 | 50000 | 1024 | 1024 |
| **epochs** | 200 | 200 | 200 | 500 | 500 | 500 | 1000 | 100 | 100 |
| **learning rate** | 0.005 | 0.01 | 0.01 | 0.01 | 0.01 | 0.002 | 0.001 | 0.001 | 0.001 |
| **weight decay** | 5e-7 | 5e-7 | 5e-7 | 0 | 0 | 0 | 0 | 0 | 0 |
| **dropout** | 0.5 | 0.5 | 0 | 0.5 | 0.5 | 0.5 | 0.5 | 0.5 | 0.5 |

**Table 8: Hyperparameters for GNNs (b)**

|  | IMDB [11] | | | DBLP [11] | | | ogbn-mag [20] | | |
|---|---|---|---|---|---|---|---|---|---|
|  | **GCN** | **SAGE** | **GAT** | **GCN** | **SAGE** | **GAT** | **RGCN** | **GraphSaint** | **ClusterGCN** |
| **layers** | 2 | 2 | 3 | 2 | 2 | 2 | 2 | 2 | 2 |
| **hidden dim** | 256 | 256 | 256 | 256 | 256 | 256 | 64 | 64 | 64 |
| **head num** | - | - | 4 | - | - | 4 | - | - | - |
| **sample size** | Full | Full | Full | Full | Full | Full | [25,20] | [30,30] | Full |
| **batch size** | Full | Full | Full | Full | Full | Full | 1024 | 20000 | 500 |
| **epochs** | 30 | 30 | 100 | 100 | 100 | 500 | 5 | 30 | 30 |
| **learning rate** | 0.001 | 0.001 | 0.001 | 0.005 | 0.005 | 0.001 | 0.01 | 0.005 | 0.005 |
| **weight decay** | 0 | 0 | 0 | 0.1 | 0.1 | 0.1 | 0 | 0 | 0 |
| **dropout** | 0.5 | 0.5 | 0.5 | 0.3 | 0.3 | 0.3 | 0.5 | 0.5 | 0.5 |

The *BCR* can be easily controlled by setting the query budgets of node selection.

*A.4.3  Downstream Classification Settings in MAG240M and ogbn-mag.* For MAG240M and ogbn-mag, graphs contain various edge types (7 types in original dataset, which will be utilized for classification). Due to graph condensation would condense different nodes together and cause type inconsistency of generated edge, we re-define the edge type via the target and background nodes connection in original and compressed graphs (4 types in total: target-target, background-background, target-background, background-target) and implement node classification test.

*A.4.4  Graph Models Settings.* In our study, we evaluate target node classification performance in downstream tasks with various GNN models. Our experiments are implemented with PyTorch 1.10.0, CUDA v12.1 on NVIDIA Quadro RTX 6000 GPU. The hyperparameters of GNN models on each dataset are shown in Table 7 and 8.

For the model with mini-batch training and node-wise based sampling settings, we report the corresponding batch size and sample size values, and the training & testing is implemented in an inductive fashion, otherwise we report them as "Full" and the node classification is conducted in transductive setting. For all graphs for node classification (i.e., original graph and compressed graphs), we follow the same hyperparameter settings for a fair comparison. Each experiment is repeated for 10 trials on all datasets except MAG240M.

*A.4.5  Varied Background Node Compression Rate Control.* To obtain the synthetic graphs with different compressed rates, we can use different condensation strategies of Graph-Skeleton with varied fetching depth $(d_1, d_2)$ for background nodes compression. Specifically, the fetching depth $(d_1, d_2)$ controls the number of fetched background nodes in the vanilla subgraph $G'$. Increasing the fetching depth results in a more comprehensive collection of information, but it also leads to a larger size of the vanilla subgraph. On the other

**Table 9: Statistics of the background nodes with various condensation strategies in synthetic skeleton graphs**

| | DGraph | | ogbn-arxiv | | ogbn-mag | | MAG240M | |
|---|---|---|---|---|---|---|---|---|
| | # Background | BCR | # Background | BCR | # Background | BCR | # Background | BCR |
| **Original** | 2,474,949 | - | 90,941 | - | 1,203,354 | - | 242,762,340 | - |
| **Skeleton-$\alpha$** | 1,044,856 | 0.422 | 46,523 | 0.511 | 800,835 | 0.665 | 5,696,810 | 0.023 |
| **Skeleton-$\beta$** | 931,842 | 0.376 | 11,217 | 0.123 | 800,835 | 0.665 | 5,696,810 | 0.023 |
| **Skeleton-$\gamma$** | 373,015 | 0.150 | 6349 | 0.069 | 479,861 | 0.398 | 2,970,934 | 0.012 |

hand, three condensation strategies control the level of condensation of redundant information in vanilla subgraph $\mathcal{G}'$, where Strategy-$\alpha$ preserves the greatest amount of the original structural and semantic information, while strategy-$\gamma$ delivers the lowest BCR. We conduct a detailed analysis to evaluate the impact of different fetching depths on compression performance in E.1, and explore the performance of three condensation strategies in Section 4.3.

*A.4.6 Settings of Three Strategies Analyses.* To investigate the condensation performance of three condensation strategies, we conduct the experiments and analyze the results in Section 4.3. During compression, we keep the fetching depths being same ($[d_1, d_2] = [2,1]$) for vanilla subgraph generation and use three strategies for condensation to generate the synthetic skeleton graphs respectively. The detailed statistics of the final synthetic skeleton graphs under three condensation strategies on four datasets in Section 4.3 are shown in Table 9.

## A.5 Discussion of Similar Performance

For ogbn-mag and MAG240M, skeleton-$\alpha$, $\beta$ present quite similar performance in Figure 7(right two). It is because that two target nodes (paper) in ogbn-mag and MAG240M are connected by either one background nodes (author) or three background nodes (author-institution-author). Thus when the node fetching distance $d_1$ is less than 4, we can only preserve the paper-author-paper edge, which can not be further condensed by strategy-$\beta$, leading to a similar condensation performance.

## A.6 Attention Settings

To further investigate how the background nodes influence the target prediction performance, we leverage the attention mechanism [33] to indicate importance of graph structure in 4.4. Specifically, we deploy the graph data on GAT for target classification and compute the importance weight of each background structural pattern. Specifically, for each pair of adjacent nodes, we calculate the attention weight between them and normalize it with softmax function.

$$e_{ij} = att(\mathbf{W}\vec{h}_i, \mathbf{W}\vec{h}_j), \qquad (7)$$

$$\alpha_{ij} = \text{softmax}_j(e_{ij}) = \frac{\exp(e_{ij})}{\sum_{k \in \mathcal{N}_i} \exp(e_{ik})}, \qquad (8)$$

where $\vec{h}_i), \vec{h}_j)$ are the hidden representations of node pair $i, j$, $W$ is a transformation matrix, $att(\cdot)$ denotes the attention computation in GAT. We use $\tilde{\alpha}_{ij} = \alpha_{ij} - \frac{1}{\deg(i)+1}$ to evaluate the importance of nodes, where $\deg(i)$ is the degree of node $i$. Note that $\tilde{\alpha}_{ij}$ could be negative, and that means the importance of this node to the target

node is lower than the average. To evaluate how the background nodes with different local structures contribute to target nodes classification, we divide the background nodes into different groups according to their structural pattern class (defined in Section 4.4) and calculate the mean $\tilde{\alpha}_{ij}$ of each group respectively.

## B PROOF OF OF PROPOSITION 1

Before the proof, let's revisit the definition of Linear message path passing on a single path (Section 3.2, Definition 3.3), then we extend the Linear message path passing function to multiple paths.

*Definition B.1 (Linear message path passing).* Given a path $p = \langle u_0, u_1, ..., u_\ell \rangle$, we define the linear single path passing functions $f_{spp}(X, p)$ which aggregates node feature starting from $u_0$ to $u_\ell$ over $p$ as:

$$X'[u_i] \leftarrow f^i_{lmp}(X'[u_{i-1}], X[u_i])W^i, i \in 1, .., \ell, \qquad (9)$$

where $X$ is the node feature matrix, $f_{lmp}(u, v)$ denotes the linear message passing operation (defined in equation 2) from $u$ to $v$, $W$ is a transformation matrix.

Given $m$ paths with same end node $K$: $P := \{p_{u_0^0, K} = \langle u_0^1, u_1^1, ..., u_*^1, K \rangle$, $p_{u_0^1, K} = \langle u_0^2, u_1^2, ..., u_*^1, K \rangle, ..., p_{u_0^m, K} = \langle u_0^m, u_1^m, ..., u_*^m, K \rangle\}$, define the linear message path passing $f_{lmpp}(X, P)$ over $m$ paths to end node $K$ as:

$$X'[K] \leftarrow \text{AGGREGATE}(X[K], \{X'[u_*^i], \forall i \in 1, ..., m\})W^K$$
$$= \text{AGGREGATE}(X[K], \{f_{spp}(X, p_{u_0^i, u_*^i})[u_*^i], \forall i \in 1, ..., m\})W^K \qquad (10)$$

PROOF. Let $X$ denotes node feature matrix, $f_{spp} = f^m_{lmp} \circ ... \circ f^2_{lmp} \circ f^1_{lmp}$ denotes a single path passing function, where $f_{lmp}$ is the linear message passing operation. Given $u, v \in \mathcal{B}$, s.t., $MSS_u = MSS_v = \{\langle \mathcal{T}, \mathcal{D} \rangle\}$. Pick one target $T \in \mathcal{T}'$ with the corresponding distance $d$, and two shortest paths $p_{u,T} = \langle u, u_1, ..., u_*, T \rangle$ and $p_{v,T} = \langle v, v_1, ..., v_*, T \rangle$ from $u, v$ to $T$ respectively, where $|p_{u,T}| = |p_{v,T}| = d$. Assuming $\{p_{u,u_*}\} \cap \{p_{v,v_*}\} = \emptyset$, let $P := \{p_{u,T}, p_{v,T}\}$, the linear message path passing over paths to $T$ is formulated as:

$$X'[T] = f_{lmpp}(X, P)[T]$$
$$= \text{AGGREGATE}(X[T], \{X'[u_*], X'[v_*]\})W^T \qquad (11)$$

$$X'[u_*] = f_{spp}(X, p_{u,u_*})[u_*]$$
$$= f^{d-2}_{lmp}(...f^1_{lmp}(X[u], X[u_1])W^1..., X[u_*])W^{d-2} \qquad (12)$$

$$X^{'}[v_*] = f_{spp}(X, p_{v,v_*})[v_*]$$
$$= f_{lmp}^{d-2}(...f_{lmp}^1(X[v], X[v_1])W^1 ..., X[v_*])W^{d-2} \tag{13}$$

Since $f_{lmp}$ is linear, the aggregated feature $X^{'}[u_*]$ and $X^{'}[v_*]$ can be decoupled as:

$$X^{'}[u_*] = \frac{X[u] \cdot W^1 \cdot ... \cdot W^{d-2}}{c^{d-2}} + \frac{X[u_1] \cdot W^1 \cdot ... \cdot W^{d-2}}{c^{d-2}}$$
$$+ \frac{X[u_2] \cdot W^2 \cdot ... \cdot W^{d-2}}{c^{d-3}} + ... + \frac{X[u_*] \cdot W^{d-2}}{c} \tag{14}$$

$$X^{'}[v_*] = \frac{X[v] \cdot W^1 \cdot ... \cdot W^{d-2}}{c^{d-2}} + \frac{X[v_1] \cdot W^1 \cdot ... \cdot W^{d-2}}{c^{d-2}}$$
$$+ \frac{X[v_2] \cdot W^2 \cdot ... \cdot W^{d-2}}{c^{d-3}} + ... + \frac{X[v_*] \cdot W^{d-2}}{c} \tag{15}$$

where $c$ is the division index, $c = 2$ for mean pooling aggregation in Eq.(2), for summation pooling aggregation $c = 1$. Combining Eq.(11) and Eq.(14),(15) we have

$$X^{'}[T] = \text{AGGREGATE}(X[T], X^{'}[u_*], X^{'}[v_*])W^T$$
$$= \frac{(X[u] + X[v]) \cdot W^1 \cdot ... \cdot W^{d-2} \cdot W^T}{c^{d-2}c^\dagger}$$
$$+ \frac{(X[u_1] + X[v_1]) \cdot W^1 \cdot ... \cdot W^{d-2} \cdot W^T}{c^{d-2}c^\dagger}$$
$$+ \frac{(X[u_2] + X[v_2]) \cdot W^2 \cdot ... \cdot W^{d-2} \cdot W^T}{c^{d-3}c^\dagger}$$
$$+ ...$$
$$+ \frac{(X[u_*] + X[v_*]) \cdot W^{d-2} \cdot W^T}{cc^\dagger}$$
$$+ \frac{(X[T]) \cdot W^T}{c^\dagger} \tag{16}$$

where $c^\dagger = 3$ for mean pooling aggregation in Eq.(11), $c^\dagger = 1$ for summation pooling aggregation. Similarly, for $\tilde{X}$ where $\tilde{X} = X$ except $\tilde{X}[u] = X[v], \tilde{X}[v] = X[u]$, we have

$$X^{'}[T] = \tilde{X}^{'}[T] \Leftrightarrow u \simeq_{\mathcal{LMPP},T} v \tag{17}$$
$$\square$$

This indicates that the aggregated information via linear message path passing is only related to the path length, but not the other nodes on the path. The proposition also holds when $\{p_{u,u_*}\} \cap \{p_{v,v_*}\} \neq \emptyset$ and each $f_{lmp}^m$ in $f_{spp} = f_{lmp}^m \circ ... \circ f_{lmp}^2 \circ f_{lmp}^1$ employs different aggregation, we omit the proof since it is similar.

## C  TIME COMPLEXITY ANALYSIS

Our method is divided into two parts, node fetching and graph condensation.

For the first part, we drop the node where the length of the shortest path is greater than $d_2$ or the length of the shortest path + the second short path is greater than $d_1$. This can be achieved through a multi-source shortest path problem (all target nodes as the source). In this algorithm, each node will be accessed at most twice, each node will be enqueued at most twice, and their adjacent

---

**Algorithm 1:** Bridging Background Node Fetching

**Input:** Graph $\mathcal{G} = (\mathcal{V}, \mathcal{E})$, target nodes $\mathcal{T} = (T_1, T_2, ..., T_n)$, $d_1$

**Output:** The bridging node set $Brid$

1  Initialize a queue $Q$ with all target nodes $\mathcal{T}$;
2  $Brid \leftarrow \emptyset$;
3  **while** $Q$ is not empty **do**
4      $u \leftarrow Q$.dequeue();
5      **foreach** $v$ is a neighbor of $u$ **do**
6          **if** edge $(u, v)$ has been accessed less than twice **then**
7              Accesse edge $(u, v)$;
            `// Note that the shortest and`
                    `2nd-shortest path are maintained in`
                    `the update process so that they`
                    `start from different target nodes.`
8              Update the shortest and 2nd-shortest path of $v$ by $u$;
9              $Q$.enqueue($v$);
10  **foreach** $u \in \mathbf{V}/\mathcal{T}$ **do**
11      **if** $u$'s length of shortest path + 2nd-shortest path $\leq d_1$ **then**
12          $Brid \leftarrow Brid \cup u$
13  **return** $Brid$

---

edges will be enumerated twice. And the time complexity of this step is $O(|E|)$. Specifically, to find shortest paths in an unweighted graphs via BFS, the complexity is $O(|E| + |V|) = O(|E|$ (assuming no isolated nodes, $|E| > |V|$) [8]. For weighted graphs, we use the same algorithm. Please note that we aim to find the minimum hop paths while not the minimum weight paths in weighted graphs, so the complexity is also $O(|E|)$.

After that, the nodes in the k-order neighborhood of each target node are accessed to calculate $PCC$. Let the average number of edges in each background node's $k$-hop ego-network be $e(k)$, then the time complexity of the above steps is $O(e(k)|V|)$. The total time complexity of the first part is $O(|E| + e(k)|V|)$.

For the second part, we propose three graph condensation methods ($-\alpha$, $-\beta$ and $-\gamma$ respectively). All these methods need to calculate the distance between each background node and the target nodes in its $k$-hop ego-network. We use the hash method to merge similar nodes, so the subsequent merging steps can be completed with $O(1)$ for each node. This makes the time complexity of the whole process determined by the complexity of previous distance calculating, that is, $O(e(k)|V|)$

The first two methods ($\alpha$, $\beta$) end after merging nodes with the same hash value, while the last method($\gamma$) requires an additional step. For Graph Condensation-$\gamma$, merging of affiliation nodes for each target node only requires traversing the neighbors of these target nodes, which means that the time complexity of this additional step is $O(|E|)$. Therefore, the time complexity of these three graph condensation methods for the second part is $O(|E| + e(k)|V|)$. To sum up, the total time complexity is $O(|E| + e(k)|V|)$.

---

**Algorithm 2:** Affiliation Background Node Fetching

**Input:** Graph $\mathcal{G} = (\mathcal{V}, \mathcal{E})$, feature matrix $X$, target nodes
$\qquad \mathcal{T} = (T_1, T_2, ..., T_n)$, $d_2, K$

**Output:** The affiliation node set $Affil$

1  Initialize an queue $Q$ with all target nodes $\mathcal{T}$;

2  Initialize the an array $dis$ with $\infty$ for background nodes and
$\quad$ 0 for target nodes;

3  $Affil \leftarrow \emptyset$;

4  $Affil' \leftarrow \emptyset$;

5  **while** $Q$ is not empty **do**

6  $\quad$ $u \leftarrow Q.$dequeue();

7  $\quad$ **foreach** $v$ is a neighbor of $u$ **do**

8  $\quad\quad$ **if** $dis[v]$ is $\infty$ **then**

9  $\quad\quad\quad$ $dis[v] \leftarrow dis[u] + 1$;

10  $\quad\quad\quad$ $Q.$enqueue($v$);

11  **foreach** $u \in \mathbf{V}/\mathcal{T}$ **do**

12  $\quad$ **if** $dis[u] \leq d_2$ **then**

13  $\quad\quad$ $Affil' \leftarrow Affil' \cup u$

14  **foreach** $u \in \mathcal{T}$ **do**

15  $\quad$ $pcc \leftarrow \emptyset$;

16  $\quad$ $affil \leftarrow \emptyset$;

17  $\quad$ **foreach** $v$ is a neighbor of $u$ **do**

18  $\quad\quad$ **if** $v \in Affil'$ **then**

19  $\quad\quad\quad$ $affil \leftarrow affil \cup v$;

20  $\quad\quad\quad$ $pcc \leftarrow pcc \cup coorelation(X_u, X_v)$;

21  $\quad\quad\quad$ sort $affil$ according to $pcc$;

22  $\quad\quad\quad$ $Affil \leftarrow Affil \cup affil[1 : K]$;

23  **return** $Affil$

---

**Algorithm 3:** Condensation-$\alpha$

**Input:** Graph $\mathcal{G} = (\mathcal{V}, \mathcal{E})$, target nodes $\mathcal{T} = (T_1, T_2, ..., T_n)$,
$\qquad$ Affiliation and Bridging node set $Affil, Brid$,
$\qquad$ ego-network's hop $k$

**Output:** Condensed skeleton graph $\mathcal{S}_\alpha$

1  Set $mask[u]$ to 0 for every node $u$;

2  **foreach** target node $u \in \mathcal{T}$ **do**

3  $\quad$ **for** $d \leftarrow 1$ to $k$ **do**

4  $\quad\quad$ $key[u, d] \leftarrow$ an unique random 256-bit integer;

5  $\quad$ **foreach** node $v$ in $u$'s $k$-hop ego-network **and**
$\quad\quad$ $v \in Affli \cup Brid$ **do**

6  $\quad\quad$ Let $d$ be the distance between $u$ and $v$;
$\quad\quad$ // $\odot$ denote Binary Exclusive Or(xor).

7  $\quad\quad$ $mask[v] \leftarrow mask[v] \odot key[u, d]$;

8  Sort the node by node's $mask$ with Radix Sort;

9  Merge nodes with the same mask into one node and get $\mathcal{S}_\alpha$;

10  **return** $\mathcal{S}_\alpha$

---

**Algorithm 4:** Condensation-$\beta$

**Input:** Graph $\mathcal{G} = (\mathcal{V}, \mathcal{E})$, target nodes $\mathcal{T} = (T_1, T_2, ..., T_n)$,
$\qquad$ Affiliation and Bridging node set $Affil, Brid$,
$\qquad$ ego-network's hop $k$

**Output:** Condensed skeleton graph $\mathcal{S}_\beta$

1  Assign each target node $u$ with an unique random 256-bit
$\quad$ integer $key[u]$;

2  Set $mask[u]$ to 0 for every node $u$;

3  **foreach** target node $u \in \mathcal{T}$ **do**

4  $\quad$ **foreach** node $v$ in $u$'s $k$-hop ego-network **with**
$\quad\quad$ $v \in Affli \cup Brid$ **do**
$\quad\quad$ // $\odot$ denote Binary Exclusive Or(xor).

5  $\quad\quad$ $mask[v] \leftarrow mask[v] \odot key[u]$;

6  Sort the node by node's $mask$ with Radix Sort;

7  Merge nodes with the same mask into one node;

8  Update edge weights by original distance information of
$\quad$ merged nodes and get $\mathcal{S}_\beta$;

9  **return** $\mathcal{S}_\beta$

---

# D  ALGORITHMS IMPLEMENTATION DETAILS

In this section, We show the detailed implementations of Graph-Skeleton. Specifically, we present the corresponding background fetching and graph condensation strategies respectively.

## D.1  Background Node Fetching

**Bridging Background Node.** We show the detailed process of bridging background node fetching in Algorithms 1. In detail, we first initialize a queue $Q$ with all target node $\mathcal{T}$ and set the initial bridging background node set $Brid$ as empty. Then we search for all accessible nodes starting from the target nodes in $Q$ and update their corresponding shortest and 2nd-shortest paths to target nodes (Line 3-9). Then we select the background nodes under our proposed fetching principle 1 of distance $d_1$ in Section 3.1 as the bridging background nodes $Brid$ (Line 10-12).

**Affiliation Background Node.** We show the detailed process of affiliation background node fetching in Algorithms 2. In detail, we search for all accessible nodes starting from the target nodes in $Q$ and obtain their corresponding distances between each other (Line 5-10). Then we select the background nodes under our proposed fetching principle 2 of distance $d_2$ in Section 3.1 as the affiliation

node $Affil'$ (Line 11-13). To control the width of fetching, we compute the Pearson correlation coefficients (PCC) between the target nodes and the their neighbors and select the background nodes with largest $K$ PCCs as the affiliation background nodes $Affil$ (Line 14-22).

## D.2  Graph Condensation

We show the detailed algorithm of graph condensation. The condensation strategy-$\alpha$ is shown in Algorithms 3. In detail, we formulate the multiple structure-set $MSS$ ($mask$ in Algorithms 3) for each background node in $Affil \cup Brid$ via its accessible target nodes and the corresponding shortest path distances (Line 2-7), then for the background nodes with same $MSS$ ($mask$), we merge them into a synthetic node (Line 8-9). For condensation strategy-$\beta$, we only

**Table 10: Sensitivity Analysis of Target Nodes Sparsity.** $r$ **denotes the target masking ratio. Columns 2-4: The averaged count of target node (ration to the total neighbors count) at** $k$**-th hop for each central target node. Columns 5-7: The test accuracy based on the target masked graph and the corresponding compressed skeleton graphs with different settings of** $d_1, d_2$**.**

| $r$ | 1st hop $N_t$ (ratio) | 2nd hop $N_t$ (ratio) | 3rd hop $N_t$ (ratio) | Original | Skeleton ($d_1 = 3, d_2 = 3$) | Skeleton ($d_1 = 2, d_2 = 1$) |
|-----|-----------------------|-----------------------|-----------------------|----------|-------------------------------|-------------------------------|
| 0.0 | 1.11 (0.123) | 3.03 (0.028) | 8.62 (0.009) | 71.02±0.45 | 71.09±0.27 | 70.84±0.33 |
| 0.1 | 0.91 (0.094) | 2.78 (0.021) | 7.54 (0.007) | 70.76±0.41 | 70.01±0.46 | 69.55±0.40 |
| 0.3 | 0.69 (0.075) | 1.69 (0.016) | 5.12 (0.005) | 70.03±0.37 | 69.76±0.35 | 68.97±0.21 |
| 0.6 | 0.28 (0.051) | 0.59 (0.006) | 1.69 (0.002) | 68.50±0.58 | 68.18±0.65 | 67.33±0.60 |
| 0.9 | 0.09 (0.011) | 0.305 (0.004) | 0.71 (0.001) | 64.89±0.55 | 64.81±0.38 | 64.03±0.46 |

**Table 11: Compression performance with varied fetching distances** $d_1, d_2$ **under Skeleton-**$\alpha$ **strategy. "Gene_t" indicates time costs of skeleton generation (graph compression), "BCR" indicates background node compression rate. The target node classification performance is evaluated by GraphSAGE.**

| Dataset | Graph | $d_1, d_2$ | Gene_t | BCR | Test Score (%) |
|---------|-------|------------|--------|-----|----------------|
| DGraph | Original | - | - | 0.00 | 78.68±0.06 |
|  | Skeleton-$\alpha$ | [3, 3] | 51s | 0.76 | 78.61±0.06 |
|  |  | [2, 2] | 36s | 0.65 | 78.56±0.06 |
|  |  | [2, 1] | 32s | 0.47 | 78.52±0.07 |
| arxiv | Original | - | - | 0.00 | 71.02±0.45 |
|  | Skeleton-$\alpha$ | [3, 3] | 35s | 0.75 | 71.09±0.27 |
|  |  | [2, 2] | 8s | 0.69 | 71.02±0.31 |
|  |  | [2, 1] | 5s | 0.48 | 70.84±0.33 |

involve the accessible target nodes of each background node in $Affil \cup Brid$ while omitting the corresponding distance information in $MSS$ of strategy-$\alpha$ (Line 1, Algorithms 4). After merging the background nodes with same $MSS$ (L) to a synthetic node (Line 8-9), we update the edge weights of condensed graph by the original distance information of merged nodes. For condensation strategy-$\gamma$, we omit the detailed pseudocode since it is similar to strategy-$\beta$ but with the last step of merging the affiliation background nodes to their corresponding target nodes using Equation 5.

# E    ADDITIONAL EXPERIMENTAL RESULTS

## E.1    Analysis of Fetching Depths

In this subsection, we conduct experiments on the ogbn-arxiv and DGraph datasets to explore the influences of node fetching depths of $d_1$ and $d_2$ to graph compression performance. We compress the graph data with varied fetching depths $d_1$, $d_2$ and use Skeleton-$\alpha$ strategy for node condensation since it preserves most background node information and is most sensitive to fetching depths.

Considering most of the popular GNN backbones contain no more than three convolution layers, we initially set the $d_1$ and $d_2$ no more than 3 (although structural connectivity beyond 3-hops cannot be guaranteed, it has no impact on information aggregation in these GNNs). The compression performance is shown in Table 11, where we report the skeleton graph $\mathcal{S}_\alpha$ generation time costs, background node compression rate ($BCR$) and target node classification performance in downstream tasks. The test scores in downstream tasks are evaluated by GraphSAGE. As we can see,

smaller fetching depths ([$d_1,d_2$]=[2,1]) notably increase the $BCR$ and reduces the compression time consumption, while presenting negligible performance degradation of target node classification in downstream tasks.

## E.2    Sensitivity Analysis of Target Nodes Sparsity

Considering the sparsity of target node within the web graphs might influence the background nodes fetching, we conduct an additional experiment based on ogbn-arxiv graph to evaluate the relationship between the target sparsity and sensitivity of $d_1, d_2$. To measure the sparsity of target nodes within the graph, we employ the count (ratio) of target nodes in the $k$-order neighborhood of each target node as the metric to measure how sparse the target nodes are. Specifically, we compute this target node sparsity metric within 3-hops for each target node on ogbn-arxiv dataset. To regulate the sparsity of the target nodes, we randomly mask a fixed rate $r$ of target nodes as background nodes within the ogbn-arxiv graph. Then we compress these graphs using skeleton-$\alpha$ with different settings of $d_1, d_2$. The results are shown in Table 10, where each row depicts the statistics and test accuracy of the graph with a target masking ratio $r$. Columns 2-4 display the averaged count of target node (ration to the total neighbors count) at $k$-th hop for each central target node, while columns 5-7 present the test accuracy based on the target masked graph and the corresponding compressed skeleton graphs with different settings of $d_1, d_2$. The initial row presents the statistics and test accuracy of the original ogbn-arxiv graph, with a masking ratio of $r = 0$.

As we can see, within the original ogbn-arxiv graph, each target node maintains an average of one target node as its immediate neighbor, three target nodes as its 2nd-order neighbors, and eight target nodes as its 3rd-order neighbors, which results in a more dispersed sparsity among target nodes. As illustrated in the table, for various target masking ratios $r$, the compressed graphs (skeleton) with $d_1 = 3, d_2 = 3$ can still well approximate the performance of the original graph. For skeleton ($d_1 = 2, d_2 = 1$), we can see that the performance gap with the skeleton ($d_1 = 3, d_2 = 3$) becomes larger with masking ration $r$ increasing (more sparse target nodes scattering). This might be due to the fact that when the target nodes are highly sparse scattering, skeleton ($d_1 = 2, d_2 = 1$) can hardly fetch the bridging nodes within $d_1$, and the fetched affiliation nodes are also much fewer than skeleton ($d_1 = 3, d_2 = 3$), leading to the performance gap. Despite that, it remains clear that the compressed graphs with different settings of $d_1, d_2$ presents close

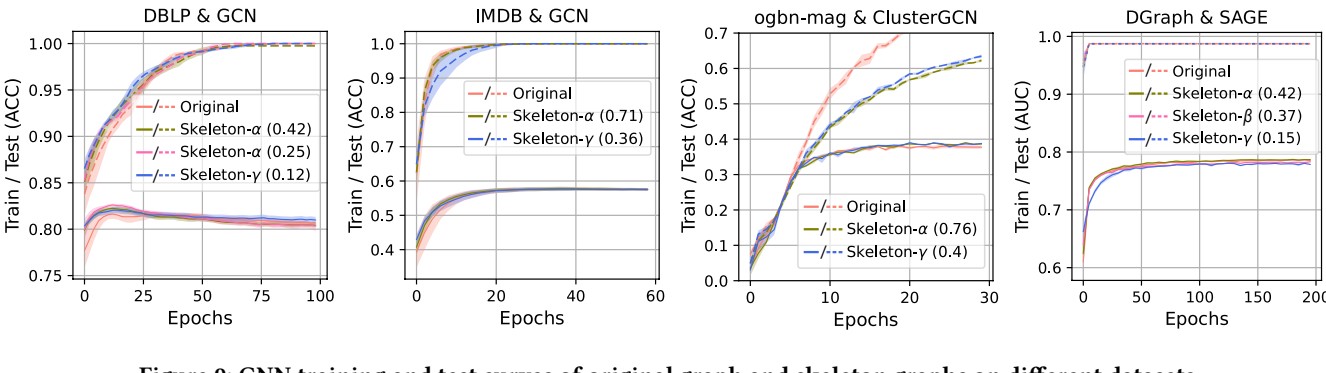

Figure 9: GNN training and test curves of original graph and skeleton graphs on different datasets

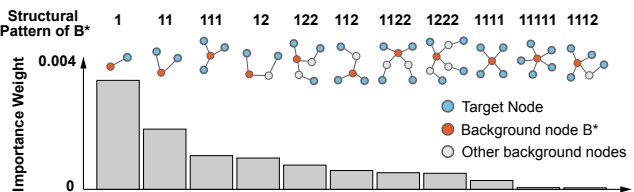

Figure 10: The importance weights of top 10 structural patterns of background nodes in $\mathcal{S}_\alpha$ of ogbn-arxiv. The corresponding structural patterns are visualized upper the bars.

node classification performance compared to the original graph, indicating that our method has good robustness to the graphs with different target nodes sparsity.

### E.3    Convergence Analysis of Skeleton Graph

To explore the convergence performance of graph model on our generated skeleton graph, we compare the training and test performance of GNNs on skeleton graphs to the original graph. The convergence curves of GNNs on four different datasets are presented in Figure 9. The *BCR* of each synthetic graph is shown in legend, the dash lines represent the training scores and the solid lines represent the testing scores. It is clear that the GNN can achieve convergence rates on the skeleton graph-$\gamma$ that are very close to those of the original graph. Similar performances are also observed in the testing score curves. This illustrates that our proposed method can well preserve the information that is essential for both model training and inference.

### E.4    Exploration of Background Nodes Influences on ogbn-arxiv

We also take ogbn-arxiv as another case to investigate how the background nodes influence the performance in the citation network. The attention coefficients of the top 10 patterns and the corresponding local structures are presented in Figure 10. Different from DGraph, the affiliation nodes deliver the most essential influence on the target prediction, indicating that the correlation information is more important for paper area prediction. This is reasonable because for the paper citation network, the direct neighbor (cited paper) is most likely to belong to the same domain as the

target node. On the other hand, the target connectivity also contributes to the task ({1,1}, {1,1,1} rank second and third respectively). This can also be well explained by the fact that the articles cite the same papers that may belong to the same field.

## F    DISCUSSION

In this paper, we focus on a common challenge in real-world applications: compressing the massive background nodes for predicting a small part of target nodes within the graph data, to ease GNNs deployment and guarantee the performance. Specifically, we propose a effective yet efficient graph compression method to compress the background nodes while maintaining the vital semantic and structural information for target nodes classification. To the best of our knowledge, this is the first work on the background nodes compression problem for target node analysis, addressing the bottleneck of data storage and graph model deployment in real world applications.

Since some studies [22, 24, 25] have been conducted for graph compression, here we emphasize the key differences and advantages of Graph-Skeleton in the following aspects: (1) our method strictly preserve the whole target nodes while restricting the the compression candidate nodes in the union of background nodes. It allows us to easily trace and analyse the information of each individual target node in the compressed graph. However, the methods mentioned above would inevitably lose some target nodes or have no ability to handle the background node compression. (2) our method is highly efficient both on space and time costs. The above studies require large memory costs during compression since complex and large parameter space for synthetic graph structure and node features generation, which greatly impedes the deployment in real-world large-scale graphs. Instead, our method uses a simple yet effective algorithm, which presents great scalability even on very large-scale graphs (MAG240M with 2.4 million nodes) while preserving comparable performance to the original graph. Besides, the time cost of our method is also very low, which only takes 5s to compress the ogbn-arxiv data with more than 0.16 million nodes (Table 11). (3) The studies mentioned above aim to generate a small sugbraph for training acceleration while still require the entire graph for inference, analysis and storage. In contrast, the compressed synthetic graph of our method extends the benefits of a reduced scale to all aspects, including inference, analysis, and storage. (4) Our method can compress the graph once for all, with good adaptability

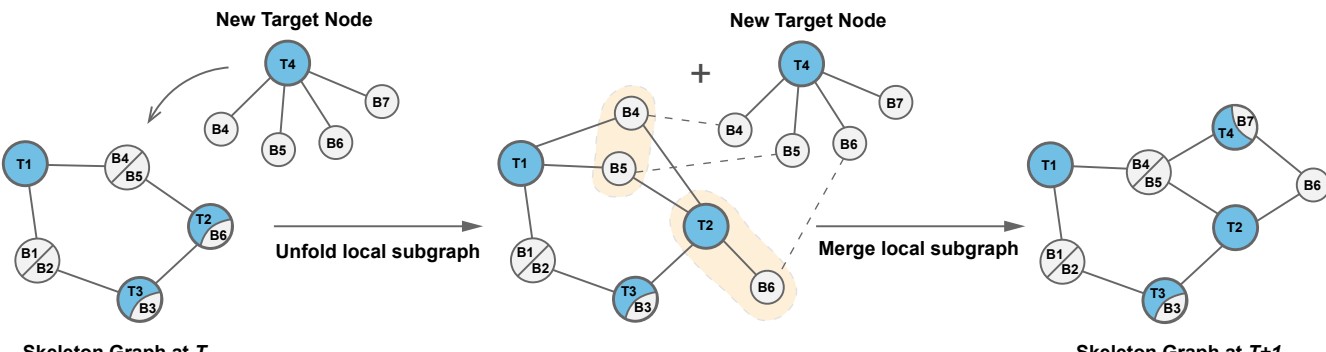

**Figure 11: A toy illustration of how the proposed Graph-Skeleton preserves the original long-range dependencies maintained by the target nodes and bridging background nodes. The fetching depths $d_1, d_2$ are set to 2 and 1 respectively. Two dashed lines represent the examples where the dependencies between the distant node pairs ($\{B1, T6\}$ and $\{T2, T5\}$, 7-hops and 4-hops away respectively) can also be preserved in the compressed skeleton graph. These preserved long-range dependencies can be well captured by the GNNs that are capable of exploring more neighborhood hops.**

**Figure 12: A toy example of how the proposed Graph-Skeleton handles the newly emerged target nodes in a dynamic evolution graphs. It incorporates the new target node into the old synthetic skeleton graph under a local subgraph condensation manner.**

to various downstream GNNs, without requiring distinct compression tailored for each individual GNN. In addition, our compressed graph can also well preserve the dependency between distant nodes, allowing deep graph models to explore multiple neighborhood hops. for which we will give detailed discussion below. (5) Our method has high flexibility on the dynamic evolution graphs. We present detail illustration in the following section.

**Long Range Dependency Preservation** Our compressed graph can also preserve the dependency between distant nodes even when the $d_1$ and $d_2$ are small. Specifically, our method only strictly restricts the hop size of affiliation background nodes into $d_2$ for each target node, but the distant dependencies maintained by the target nodes and bridging background nodes can be well preserved. We present a toy example for better illustration in Figure 1 in our uploaded PDF in global response. In Figure 1, the skeleton graph is compressed with the settings of $d_1, d_2 = [2, 1]$. The two dashed magenta lines indicate the potentially important dependencies between distant node pairs ($\{B1, T6\}$ and $\{T2, T5\}$, 7-hops and 4-hops away respectively) for target nodes classification. As we can see, these long-range dependencies can also be effectively preserved in our compressed graph, allowing graph models to explore multiple neighborhood hops.

**Flexibility on Dynamic Evolution Graphs**

A lot of domains now present highly dynamic data that exhibit complex temporal properties. For example, new users (nodes) and relations (edges) will continue to appear in social media platforms like Facebook and Instagram. The dynamic evolution of a graph would continuously introducing new features and labels that are out of distribution (OoD) to the original graph. This requires the continuous updates of the compressed graph so that it contains the latest information for effective model training and target nodes analysis.

Our proposed Skeleton-Graph can also be well extended to the dynamic evolution graphs. To illustrate this, we provide a toy example in Figure12 where Graph-Skeleton can easily updates the compressed graph with a newly emerged target node $T_4$ at time $T$. In details, when a new target node $T_4$ emerging connected with some neighbors ($B_4, B_5, B_6, B_7$), we first unfold the local entities (synthetic nodes) associated with $T_4$' neighbors, and then condense these unfolded nodes under the Skeleton-Graph framework to generate new entities (synthetic nodes). In this way, we only need to update the local subgraph associated to the new emerged target nodes and their neighbors while not have to update the entire original skeleton graph. This allows us to easily incorporates the new

 

information into the compressed graph in an continuous manner. In contrast, the other graph compression methods [22, 24, 25] in last section have to rebuild the compressed graph from scratch to incorporates new information, which largely impedes the implementation in real-word dynamic evolution graphs.

**Limitations and Future Works**

The limitations of our method are summarized as follows: (1) Due to the lossy compression, our method would inevitably cause the information loss and performance decline on some datasets. There is a trade-off between the graph-size and information integrity. (2) The final graph size and *BCR* of our condensed skeleton graph also depend on the target node size in the original graph data since we aim to compress the background nodes while preserving all the target nodes. For the applications with most nodes are target nodes within the graph, the compression rate would be limited. (3) The condensation performance of three proposed condensation strategies varies from different datasets with different graphs structures. For example, the condensation stratigy-$\beta$ are not that effective to condense the datasets of ogbn-mag and MAG240M (detailed illustration in A.5).

## G CODE DEMO

https://github.com/GraphSkeleton/WWW24_GraphSkeleton.git

