# OpenReview forum: "Graph-Skeleton: Less than 2% Nodes are Sufficient to Represent Billion-Scale Graph"
_ACM.org/TheWebConf/2024/Conference — TheWebConf24 Oral_

### Official Review · Reviewer_QryU · 2023-11-06

**Novelty:** 6
**Technical Quality:** 6

**Review:**

This paper studies the problem of compressing large graph in order to facilitate the execution of ML learning tasks and reduce storage and resource requirements. It addresses the specific case where only a subset of nodes need to be analyzed (e.g., nodes to classify), these nodes are called target nodes in this paper, whereas the remaining of nodes are called background nodes. The proposed compression methods, called Skeleton, keeps all the target nodes but compresses the background nodes to remove redundancy and keep only useful information for analyzing target nodes. Before presenting the approach, the paper starts by performing an empirical analysis on two graphs, to identify the impact of removing some types of nodes and edges. This study made it possible to identify edges that are not useful for the main task, and that can be removed, but also, it has identified the edges whose removal reduces significantly the accuracy of the main task. Based on these observations, the compression approach has been designed. It starts by fetching only nodes and edges that are useful for the main task. Then, it introduces three compression strategies that can be used to compress the remaining edges and nodes: \alpha, \beta, and \gamma. The \alpha compression is the one that keeps the most of relevant information, whereas \gamma compression provides the best compression ratio. In the Experiments section, the proposed approach has been compared with 9 competitive methods in 6 datasets. Each of these methods has been used to compress graphs, and then a GNN classification task is performed on the resulting compressed graphs. Experiments show that the graph compressed with the proposed approach (Skeleton) provides the best AUC and ACC scores. Empirical evaluation also studies the compression ratio of the different used strategies, and provides a data exploration section to identify the background nodes that are the most important for the prediction task applied to target nodes.

The paper has many strong points:
- The problem has been well motivated.
- The empirical study in Section 2 is very interesting, and provides a good support to motivate the choices that have been made in the Skeleton method
- The method has been well presented, with illustrative examples that explain important parts
- Empirical evaluation provides positive results
- The source code has been made available, and the supplementary materials give significant details about the experimental setting

There are some weaknesses that can be improved, or points that are missing:
- Quantitative study of execution time. The paper does not show the time required to perform the Skeleton compression, in order to compare it with the time required with existing methods. Also, this study can show the impact of dimensions (nb nodes and nb edges) on the execution time.
- It would have been interesting to show the time required to train the GNN models in the original graphs vs in the compressed graphs
- Table 3 shows the data storage of Skeleton, but not for the other competetive approaches
- While the method is really interesting, it does not provide any guarantee about the quantity of information (or the %) that will be kept in the compressed graph. This is not a critical weakness, but it would have been nice to have some guarantees. In fact, this can help to convince that the approach would work well even in graphs other than the ones considered in the experiments.


Typo:
Line 314: Figure 11 -> Do you mean Figure 2?
Line 627: Finlay -> Finally

**Questions:**

- Coud you provide the execution time required to compute the compressed graphs for Skeleton and competitive methods?
- could you provide the time required to train/infer under the original graph vs the compressed graph?

**Ethics Review Description:**

No issue

**Reviewer Confidence:**

3: The reviewer is confident but not certain that the evaluation is correct

**Scope:**

4: The work is relevant to the Web and to the track, and is of broad interest to the community

---

### Official Review · Reviewer_pEGe · 2023-11-19

**Novelty:** 4
**Technical Quality:** 4

**Review:**

Summary of contribution

In this paper, the authors target on the background nodes compression problem on large-scale graphs, so as to facilitate node classification task. Specifically, they propose a Graph-Skeleton to first fetche the useful background nodes from massive web graph and then perform background node condensation (with three strategies) to eliminate information redundancy. Experiments are conducted to demonstrate their method’s effectiveness in both accuracy and storage overhead.

Strong points

S1. This paper tackles an important and interesting problem that is critical to node classification task on large-scale graphs.

S2. The proposed Graph-Skeleton is reasonable.

S3. Experimental study is quite solid.

Weak points

W1. I am not convinced with Sec-2, more details of empirical analysis should be clarified. [D1,D2].

W2. I confused with several parts of the experimental study, which should be discussed clearly, see [D3,D4,D5].


Detailed comments

D1. The authors perform empirical analysis by employing three GNN models with three aggregations on DGraph (million-scale) and ogbn-arxiv (only with 0.16M nodes). I am not convinced that a general conclusion or statement could be obtained based on such a non-systematic setup. (a) Since the paper targets on background nodes compression on large-scale graphs, the empirical analysis should be performed on at least one billion-scale graph. More precisely, a better choice is to classify the existing graphs into several categories according to their types, difficulty, and size, and then take at least one graph from each category for analysis. (2) For node classification task, besides GraphSAGE, GAT, GIN, many Transformer-based models should be considered in empirical analysis too, e.g., NodeFormer, NAGphormer. Similarly, a better choice is to divide existing models into several categories and select at least one model from each category to perform empirical analysis.

D2. In empirical analysis, after cutting a bunch of edges, you get a new graph. How do you get the new AUC results on this new graph (i.e., results shown in Fig.1), running the trained classification model (with original parameters) or training a new one for this new graph? If you use the trained model, the results obviously would be bad. It’s more reasonable to retrain the model for the new graph. If you do so, I am interested in why the retrained model is invalid for this new graph. Note that in NodeFormer, the authors claim that the features contribute more to the classification results than structure, so if you retrain a model for a new graph, it’s reasonable that the features of target nodes would still contribute a lot to the classification results.

D3. Since GCN’s training relies on the size of adjacent matrix, the large the graph, the more the resource required for training. Training a GCN model on a million-scale graph is quite time-consuming, I was wondering the hardware settings of this paper’s experimental study (it’s better to provide them at the beginning of Sec-4). Why NAGphormer is only performed on DGraph, readers may expect to see NAGphormer’s results on all graphs. Besides NAGphormer, NodeFormer should be considered either.

D4. Since BCR is not a user-defined parameter to control the procedure of graph Condensation, I am confusing that how you can set a fixed BCR to get results as shown in Fig.6?

D5. Readers may expect to see the efficiency results of node compression, e.g., runtime of Graph-skeleton.

**Questions:**

Questions

Q1. Please justify the rationality of the empirical analysis.

Q2. How to evaluate the performance with fixed BCR?

Q3. Other issues mentioned in D1-D5.

**Reviewer Confidence:**

3: The reviewer is confident but not certain that the evaluation is correct

**Scope:**

3: The work is somewhat relevant to the Web and to the track, and is of narrow interest to a sub-community

---

### Official Review · Reviewer_hbEq · 2023-11-20

**Novelty:** 5
**Technical Quality:** 5

**Review:**

The paper considers an interesting problem: given a graph (with node features) with target nodes (for which the classification is to be performed) and background nodes (non-target nodes), how to compress the background nodes while preserving the node classification effectiveness? An early evaluation gives interesting insights about the nature of necessary and redundant background nodes. Building on those, authors design three effective techniques to compress background nodes and evaluate them on large scale graphs.

+ It's a well-executed, well-written work on an interesting problem. The focus on practical GNN usage (target vs. background) is especially interesting.

- One suggestion to better put this work in context is that target-background setup can also be interpreted as a heterogeneous network. The graphs in Table 2 basically have different types of nodes and one node type is chosen as the target. Having said that, it'd be worthwhile to check heterogeneous GNN literature and whether any such compression methods exist.

- Title is a bit misleading as it gives the impression that 2% of nodes is sufficient for *any* purpose. Actually, this is only true for node classification task and when there is a distinction between target/background nodes. Also, the "billion-scale" graph has 0.24B nodes so it's not actually true to call it that way.

- In experiments, what d1 and d2 values are used? Is there any guidance on how to select those for a new graph?

- A minor suggestion is to put a horizontal line in Fig 1 to show the original baseline (i.e., from where the green line cuts the y-axis). Also, last paragraph of Sec 3 refers to Fig 11, but it needs to be Fig 2.

**Questions:**

Please see above.

**Reviewer Confidence:**

3: The reviewer is confident but not certain that the evaluation is correct

**Scope:**

4: The work is relevant to the Web and to the track, and is of broad interest to the community

---

### Official Review · Reviewer_pmHa · 2023-11-23

**Novelty:** 5
**Technical Quality:** 5

**Review:**

The paper introduces a graph compression algorithm designed for efficient node classification. The key ideas include (a) dividing nodes into target nodes (those requiring class inference) and background nodes, and (b) merging structurally similar background nodes. Experimental results indicate that, in many cases, node classification performance remains almost unchanged even after compression.

S1. Complex mathematical concepts are effectively conveyed through examples and figures.

S2. The proposed idea, specifically merging structurally similar background nodes, is compelling.

S3. Surprisingly, node classification performance often remains nearly unaffected after compression.

W1. In many datasets, the size of the reduced graph remains substantial because compression exclusively targets background nodes. This is in contrast with other graph condensation methods, which offer more control over the compressed size.

W2. An assessment of the running time and memory usage of the proposed compression algorithm is needed, along with a comparison of the compression cost against the benefits it provides.

W3. It would be beneficial to measure the running time and GPU memory usage of GNN training with and without compression to demonstrate the actual advantages. Empirical comparisons with other scalable GNN approaches, such as sampling- and clustering-based approaches, can also be considered.

W4. The GNNs utilized are designed for homogeneous graphs, whereas the input graphs are heterogeneous.

W5. Due to the aforementioned weaknesses (W1-W4), assessing the practical significance of the proposed method is challenging.

W6. All algorithmic details are provided only in the appendix.

**Questions:**

Q1. (W2) Could you provide the exact running time and memory requirements of the proposed compression algorithm? Please compare the compression cost and the benefits it brings.

Q2. (W3) What is the impact of compression on the training time and GPU memory usage of GNNs?

Q3. (W4) Could you explain the adaptation of GNNs designed for homogeneous graphs to heterogeneous graphs? How does compression affect the node classification performance of GNNs designed for heterogeneous graphs?

**Reviewer Confidence:**

3: The reviewer is confident but not certain that the evaluation is correct

**Scope:**

4: The work is relevant to the Web and to the track, and is of broad interest to the community

---

### Decision · Program_Chairs · 2024-01-22

**Decision:**

Accept (Oral)

**Comment:**

The paper makes a tangible contribution to graph representation and generated consensus among reviewers.